# CIRCUITS, FEATURES, AND HEURISTICS IN MOLECULAR TRANSFORMERS

## ABSTRACT

Transformers generate valid and diverse chemical structures, but little is known about the mechanisms that enable these models to capture the rules of molecular representation. We present a mechanistic analysis of autoregressive transformers trained on drug-like small molecules to reveal the computational structure underlying their capabilities across multiple levels of abstraction. We identify computational patterns consistent with low-level syntactic parsing and more abstract chemical validity constraints. Using sparse autoencoders (SAEs), we extract feature dictionaries associated with chemically relevant activation patterns. We validate our findings on downstream tasks and find that mechanistic insights can translate to predictive performance in various practical settings.

## 1 INTRODUCTION

Designing molecules that satisfy multiple pharmacological and physicochemical constraints is a core challenge in drug development. While many architectures directly encode chemical invariants, adaptations of sequential architectures such as transformers (Vaswani et al., 2017; Radford et al., 2019) must induce these rules purely from data. Despite the lack of inductive bias, these systems have proven effective for *de novo* design (Honda et al., 2019; Chithrananda et al., 2020; Bagal et al., 2022; Ross et al., 2022). Such flexibility allows transformers to model complex distributions, making them popular for generative tasks where exploring specific regions of the computationally intractable chemical space is required.

The simplicity of transformers also makes them opaque. Seminal work on molecular transformers identified patterns related to syntactic and chemical validity (Bagal et al., 2022), yet a direct understanding of the processes involved in maintaining molecular representation is lacking. Modeling structured dependencies is a subject of active investigation in the broader transformer interpretability community (Hahn, 2020; Ebrahimi et al., 2020; Weiss et al., 2021; Yao et al., 2023), and distinguishing whether molecular transformers rely solely on memorization or have induced similar generalized algorithms for aspects of chemical validity is critical for applications in the life sciences.

This work presents a mechanistic analysis of an autoregressive transformer trained on commercially available drug-like molecules. We perform experiments to locate computational units that facilitate molecular generation.

The results presented show that molecular transformers can develop a number of specialized mechanisms involved in maintaining syntactic and chemical validity during inference. Using dictionary learning, we obtain human-understandable patterns that correspond to various chemical substructures. Performance on downstream tasks suggests that sparse activations produce useful features for molecular property prediction.

### 1.1 CONTRIBUTIONS

We train an autoregressive transformer on millions of molecules and conduct a mechanistic analysis spanning multiple levels of abstraction. We summarize our main findings below.

1. ***Ring and Branch Circuits***: We identify attention heads that implement matching of SMILES grammar elements, including a multi-head circuit for ring closures and a specialized branch-

balancing head, and show via targeted ablations that some of these heads are load-bearing for validity (Section 3.1).

2. *Valence Tracking*: We find a distributed linear representation of valence capacity in the residual stream and demonstrate that interventions along this direction monotonically modulate bond-order predictions at decision points in a chemically consistent way (Section 3.2).

3. *Chemical Features*: Using SAEs, we extract sparse feature dictionaries that align with chemically meaningful activation patterns and develop a fragment-screening pipeline that links latents to functional groups with reduced manual inspection (Section 4).

4. *Practical Applications*: We show that SAE-derived features improve transformer embeddings on several property prediction tasks and are competitive with, and often complementary to, ECFP fingerprints and supervised baselines (Section 5.1). We inject a small number of SAE features into transformer activations at inference resulting in increased structural similarity, often steering samples towards desired regions of chemical space (Section 5.2).

## 2 PRELIMINARIES

### 2.1 BACKGROUND

**Molecular Representation.** Transformers require a consistent, machine-readable representation of chemical structures. We employ the Simplified Molecular-Input Line-Entry System (SMILES) (Weininger, 1988), which encodes the complete molecular topology as ASCII strings through a formal grammar. Although a number of popular alternatives exist (Krenn et al., 2020; Gilmer et al., 2017; Axelrod & Gomez-Bombarelli, 2022), SMILES offers a balance of expressivity and out-of-the-box transformer compatibility, and has widespread adoption in chemoinformatics. Specific SMILES patterns directly correspond to recognizable chemical substructures, making both automated annotation and human understanding easier.

**Sparse Autoencoders.** Sparse autoencoders offer a promising approach to disentangling the internal representations learned by sequential models, addressing the challenge of polysemanticity, where neurons encode multiple semantically distinct concepts simultaneously. Unlike regular autoencoders, SAEs produce activation patterns where most values are near-zero, with only a few strongly activated neurons.

Formally, a sparse autoencoder consists of an encoder $E_\phi$ and a decoder $D_\theta$ trained to reconstruct input activations $x \in \mathbb{R}^n$ from a transformer layer while enforcing sparsity in the encoded representation $z = E_\phi(x) \in \mathbb{R}^m$, where $m > n$. The loss function usually combines reconstruction error and a sparsity penalty:

$$\mathcal{L}(\theta, \phi) = \mathcal{L}_{\text{recon}}(\theta, \phi) + \lambda \mathcal{L}_{\text{sparse}}(\phi)$$

The sparsity constraint is often imposed through L1 regularization:

$$\mathcal{L}_{\text{sparse}}(\phi) = \frac{1}{N} \sum_{i=1}^{N} \|E_\phi(x_i)\|_1$$

In this study, we utilize a SAE architecture similar to Bricken et al. (2023), consisting of an encoder with a Rectified Linear Unit (ReLU) activation and decoder-bias recentering, such that $z = \text{ReLU}(W_e(x - b_d))$, and a linear decoder. While newer alternatives improve upon this baseline, we find that standard ReLU SAEs already produce high-quality feature dictionaries for molecular transformers and adopt this simpler approach for our experiments.

### 2.2 EXPERIMENTAL SETUP

**Dataset and Tokenization.** We pretrain on a filtered, lead-like subset of ZINC20 containing $\sim$300M commercially available small molecules, using scaffold-based splits to avoid leakage between train and test sets (see Section F.3). Molecules are encoded as canonical SMILES and tokenized with the chemistry-aware regular expression of Schwaller et al. (2019).

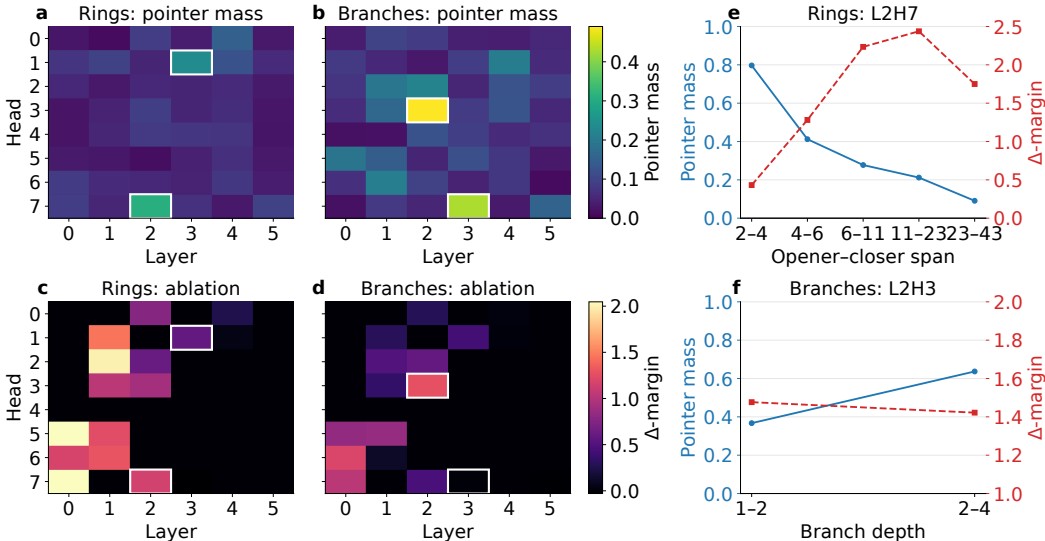

Figure 1: **Syntax Circuits in Molecular Transformers.** We find attention heads involved in pairing ring digits and balancing branch parentheses in SMILES. **(Left)** Heatmaps showing pointer mass (a-b) and the causal impact of ablation ($\Delta$-margin, c-d) for ring and branch syntax across all layers and heads. White boxes highlight the top two heads by pointer mass. **(Right)** Performance of the top specialized heads for rings (e) and parentheses (f) as a function of increasing syntactic difficulty, showing both pointer mass and $\Delta$-margin. The L2H7 head consistently identifies correct ring openers and activates more intensely at larger distances. L2H3 points at opening parentheses at branch closures.

**Transformer Model.** Unless noted otherwise, we analyze a 6-layer, 8-head decoder-only transformer (25M parameters) trained with a standard causal language modeling objective on the SMILES corpus. The model uses RMSNorm, SwiGLU feed-forward blocks, and rotary positional embeddings (Zhang & Sennrich, 2019; Shazeer, 2020; Su et al., 2024). Full architectural and training details, including optimizer and schedule, are reported in Section F.1.

**Autoencoder Training.** We train ReLU sparse autoencoders on the post-MLP residual stream activations of each layer using $10^9$ tokens, corresponding to approximately 25M sample molecules. We use three expansion factors: 4x, 8x, and 16x, corresponding to latent dimensions 2048, 4096, and 8192, respectively. Further details can be found in Section F.2.

## 3 FUNDAMENTAL CIRCUITS

### 3.1 SYNTAX: RINGS AND BRANCHES

SMILES uses specific syntactic rules for representing molecular structures: branches are enclosed in parentheses (e.g., CC(C)C for 2-methylpropane), and rings use numeric labels at connection points (e.g., C1CCCCC1 for cyclohexane). These constraints require matching opening and closing tokens across potentially long sequences, creating structured dependency challenges. To investigate how models accomplish this, we use three complementary metrics to find relevant circuits and distinguish between identification and causal execution:

1. ***Pointer Mass:*** Mean attention probability a head assigns to the correct opening token (e.g., the digit '1' in 'C1...'). For a set of grammar events $E = \{(b, i_{\text{close}}, j_{\text{open}})\}$, where each event represents a closing token at position $i$ in batch $b$ that should point to its opener at position $j$, we compute:

$$\text{Pointer Mass}(h) = \frac{1}{|E|} \sum_{(b,i,j) \in E} A_{b,i,h}[j],$$

where $A_{b,i,h}[j]$ is the attention probability that head $h$ at position $i$ assigns to position $j$. We use pointer mass to determine which heads most consistently attend to (i.e., locate) the correct source token.

2. *Event Specificity (ES):* The drop in the correct token's logit margin when the attention head is ablated, relative to control tokens, quantifying the head's local influence on the prediction. Also denoted as $\Delta$-margin.

3. *Ablation Validity:* The percentage of valid molecules generated when the head is removed, evaluating the head's global necessity for producing correct SMILES outputs.

**The Ring Circuit.** Our results in Table 1 show that ring digit matching involves two complementary attention heads with separate roles. Head **L2H7** acts as a pointer, dedicating 30.7% of its attention mass to the correct opening ring digit that tracks the open state. However, ablating this head only reduces generation validity by 10.7% compared to random ablations, suggesting the model has additional means to attend to digits.

In contrast, Head **L1H2** behaves as a *writer* (causally driving output logits). It allocates little mass to openers but exerts a massive causal influence on the output logits (Event Specificity $\approx 4.98$). Ablating this head is catastrophic, dropping mean validity to 25.4% (a decrease of 64.9%). While random single-head ablations maintain high validity (90.3%), the loss of the writer breaks the model's ability to enforce syntactic constraints, suggesting that L1H2 is a non-redundant, load-bearing component of the circuit.

As L2 pointers do not drive L1 activations in the same inference step, the findings point to a functional dissociation rather than a direct downstream dependency, where the transformer may use L1H2 for heuristic execution and L2 heads for state tracking.

**The Branch Circuit.** We find that the mechanism for branch balancing concentrates on a single attention head. Head **L2H3** consistently emerges as the sole facilitator, exhibiting both the highest pointer mass (49.0%) and significant specificity. Ablating it reduces validity to 63.7%, a significant impairment compared to controls, but less catastrophic than the head involved in ring matching. These results suggest that branch depth tracking relies on a more compact circuit where memory retrieval and execution are coupled.

| Circuit | Head | Role | Pointer Mass | Causal ES | Ablation Validity |
|---------|------|------|--------------|-----------|-------------------|
| Ring Closure | L2H7 | Pointer | **0.307** | 0.51 | 79.6% |
|  | L1H2 | Writer | 0.033 | **4.98** | **25.4%** |
| Branching | L2H3 | Hybrid | 0.490 | 0.58 | 63.7% |
| Control | Avg. [a] | Random | – | – | 90.3% |

[a] Computed from 10 randomly selected attention heads, each ablated individually. Heads in L0 are excluded from controls.

Table 1: **Functional Dissociation in Syntax Circuits.** The Ring Circuit shows a division of labor: L2H7 tracks the opener, but L1H2 is the critical bottleneck for execution. L2H3 acts as a standalone circuit for branches with a clear, but less catastrophic impact on validity. Ablations performed on 10K generated samples (Baseline validity $\sim 100\%$).

## 3.2 CHEMISTRY: VALENCE BUDGETING

We investigate whether the model stores information about *valence capacity*, i.e., the remaining bonding capacity of an atom. For each atom at each position, we compute how many more bonds it can form based on its element type, charge, and bonds already formed. We train separate multinomial logistic regression probes on the residual stream activations of each layer to predict the remaining valence capacity (integers 0–4).

**Probe Accuracy.** Linear probing achieves very high performance from Layer 1 onward and peaks at L3, indicating that a linearly readable valence budget is present in the residual stream by mid-

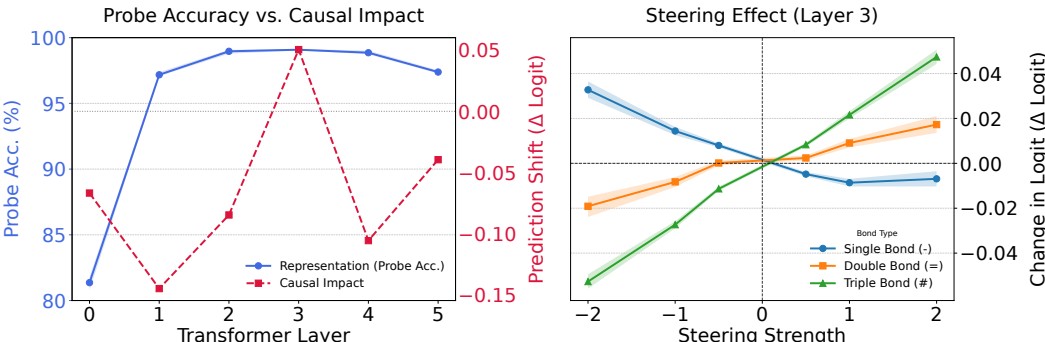

Figure 2: **Valence Capacity is Linearly Decodable.** **(Left)** Layer-wise localization. Blue line shows linear probe accuracy for predicting remaining valence. Dashed red line shows the aggregate causal effect of the valence vector on bond logits. Both metrics peak at Layer 3, indicating this layer contains the most information about valence. **(Right)** Layer 3 Steering. We intervene on the residual stream ($x \leftarrow x + \alpha\hat{w}$) at decision tokens. Increasing steering intensity shifts probability mass from single bonds ($-$) to higher-order bonds ($=$, $\#$). Shaded bands indicate 95% bootstrap confidence intervals ($N = 1000$).

stack. The probe is highly stable, achieving an accuracy of 99.08% (95% CI: $[99.03, 99.14]$), suggesting that valence constraints are encoded deterministically in the residual stream.

**Modulation.** We extract the specific valence direction $\hat{w}$ from the probe trained at Layer 3. To test whether this direction is directly linked to model predictions, we intervene by adding $\alpha \cdot \hat{w}$ to the residual stream and measure the shift in output logits for bond tokens. The intervention produces a consistent, statistically significant shift in bond probabilities. As shown in Figure 2, increasing $\alpha$ suppresses single bonds while boosting higher-order bonds.

These observations suggest the model encodes a high valence potential, regulating its willingness to predict higher-order bond tokens when the chemical context allows. The number of discrete decision flips remains low despite the direct modulation effect, indicating that the transformer can maintain validity even when the residual stream is perturbed.

## 4  CHEMICAL FEATURES

A central challenge in applying sparse autoencoders to molecules is identifying which of the thousands of learned features correspond to chemically meaningful concepts. We therefore first assess the basic quality and robustness of the SAE representations, and then describe how we identify chemistry-related features for closer inspection.

### 4.1  ROBUSTNESS AND FIDELITY

**Reconstruction.** We perform inference-time interventions by replacing the original residual-stream activations with decoded SAE embeddings to assess how well sparsified features preserve the base model's learned distribution. As detailed in Section E.1, reconstruction fidelity improves with dictionary size, with 16x SAEs achieving minimal cross-entropy loss ($\Delta$CE $< 0.02$ in L1) and maintaining near-perfect generation validity ($> 97\%$) across all layers.

**Augmentation Robustness.** To characterize SAE behavior under distribution shifts, we evaluate feature robustness using SMILES augmentation. Since only canonical molecular strings were used during pretraining, non-canonical (augmented) examples constitute an out-of-distribution shift in sequence traversal order while representing chemically identical structures. We pass the paired sequences through SAEs and compute cosine similarity to measure stability regarding activation magnitude, and Jaccard similarity to assess the stability of the active feature set. The results, shown in Figure 3 and Table 4, reveal a clear distinction.

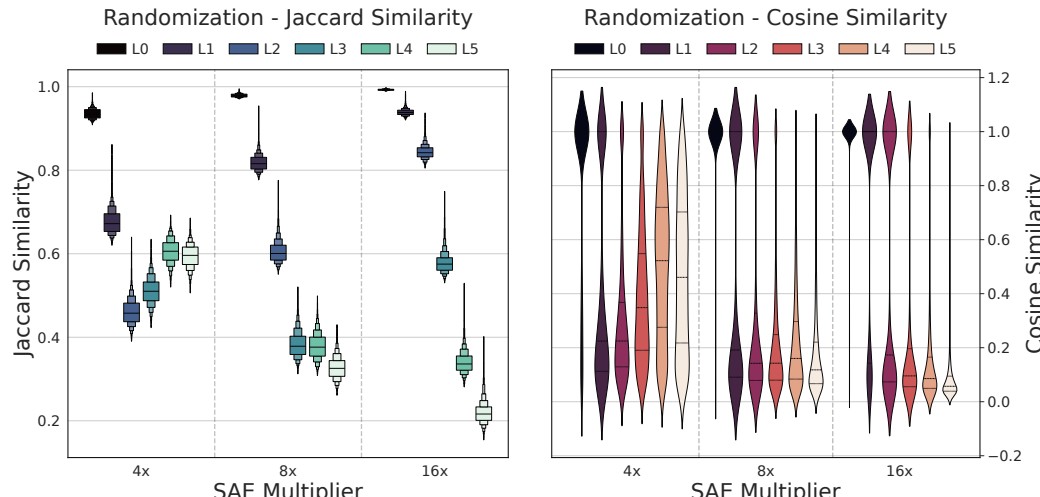

Figure 3: **Feature Robustness is Layer-dependent. (Left)** Distribution of Jaccard Similarity measuring feature set identity across layers for different SAE dictionary sizes. **(Right)** Distribution of cosine similarity scores measuring fluctuations in activation magnitude. Early layers (L0–L1) show high syntax invariance, while deeper layers show increased variance consistent with autoregressive path dependency, where activation magnitudes may fluctuate, but a significant share of features remains active. As dictionary size grows, SAEs exhibit increasingly permutation-dependent behavior.

- *Local Invariance:* Early layers exhibit high robustness across both metrics (L0 Jaccard: $\approx 0.97$, Cosine: $\approx 0.94$). This provides a clear indication that the recognition of atoms, bonds, and immediate neighbors is robust to changes in the string's starting point or direction.

- *Path Dependence:* In deeper layers, feature stability diverges. Cosine similarity drops significantly and Jaccard similarity shows a notable but less dramatic decrease, where a portion of active features retain their identity even as activation magnitudes fluctuate.

These findings suggest that SAEs successfully extract order-independent chemical entities in early layers, while the majority of features in deeper layers are more closely aligned with the models' inference process. Observed Jaccard scores indicate that SAE features can track a fraction of semantic concepts even with permuted SMILES ordering, while activation levels may vary with context. Described results suggest that wider SAEs allocate more capacity to position-dependent representations, thus are less robust to augmentation.

**Feature Universality.** A potential concern is that learned features may be artifacts of a specific SAE training run and may not be reproducible even under marginally different circumstances. We tested this by measuring the cosine alignment of features between SAEs of different sizes (4x → 16x). We find high alignment in early layers (Mean MCS $> 0.92$), suggesting different SAEs develop an overlapping basis for low-level chemistry. Deeper layers show high alignment but decreased recovery, similar to phenomena in which broad concepts are refined into granular subtypes. For cross-SAE stability statistics and further details, we point the reader to Section E.2.

### 4.2 SUBSTRUCTURE SCREENING

In the case of chemical language models, feature annotation challenges are amplified by the complexity of molecular representation. Manual feature inspection is prohibitively time-consuming and requires extensive domain expertise. To address this bottleneck, we present an automated screening process that effectively identifies SAE features likely to encode specific structural patterns in SAE activations.

We leverage the relationship between SMARTS patterns – a widely used chemical query language – and SAE feature activations to identify candidates for mechanistic investigation. The process

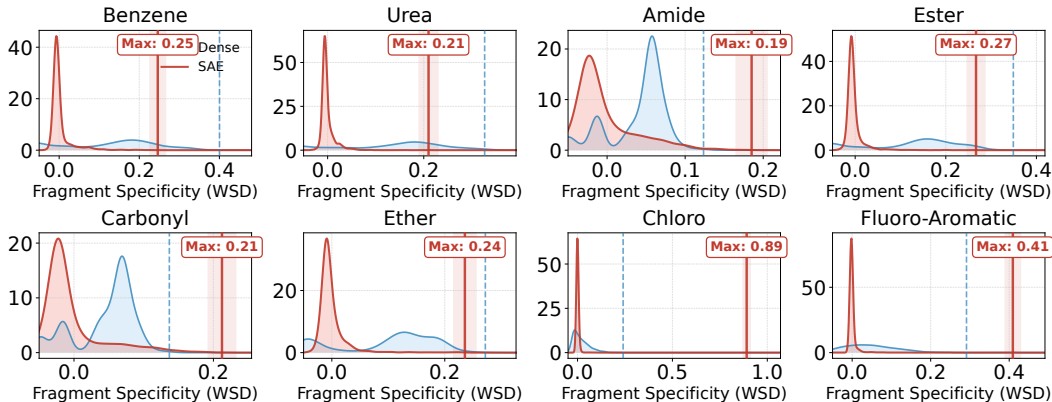

Figure 4: **Sparse Dictionaries Contain Fragment Features.** Plots show kernel density estimates of fragment-based specificity for common substructures in Layer 3 Dense MLP **(blue)** and sparse SAE representations **(red)**. The distributions highlight two complementary effects of sparsification: (i) a large mass of near-zero specificity values indicating that most SAE features remain silent or weakly responsive outside of their preferred contexts, and (ii) a distinct high-specificity tail showing that a small number of SAE neurons act as strong detectors for chemically meaningful substructures. In contrast, dense residual representations exhibit smoother, more entangled activation patterns with no comparably sharp fragment-aligned features.

consists of four stages: (1) fragment detection using compiled SMARTS patterns, (2) precise token-level attribution to map chemical fragments to transformer input tokens, (3) statistical scoring to quantify feature-fragment associations, and (4) ranking to prioritize candidates for validation.

**Intra-fragment Activations.**    To find features that activate on relevant substructures, we measure the difference in activation between fragment-containing and fragment-free positions in individual sequences. To achieve this, a coverage-weighted standardized contrast (WSD) is computed for each SMARTS pattern and autoencoder feature. Given a molecular string and a set of substructure queries, we map all matches to the corresponding token positions in the transformer's input sequence. Then, for a feature $k$ in sequence $b$, we compute:

$$\text{WSD}_{b,k} = \frac{\mu_{b,k}^{\text{in}} - \mu_{b,k}^{\text{out}}}{\sigma_{b,k}^{\text{all}} + \varepsilon} \cdot \sqrt{p_b(1 - p_b)} \tag{1}$$

where $\mu_{b,k}^{\text{in}}$ and $\mu_{b,k}^{\text{out}}$ are the mean activations on fragment and non-fragment tokens respectively, $\sigma_{b,k}^{\text{all}}$ is the standard deviation across all valid tokens, and $p_b$ is the fragment coverage. The coverage weighting term $\sqrt{p_b(1 - p_b)}$ focuses on cases where the discrimination is most informative. We rank features by WSD to flag relevant SAE neurons for manual analysis.

Across a set of common functional groups as shown in Figure 4, sparse autoencoders consistently produce a small number of highly selective, chemically meaningful features. While dense residual activations contain fragment-related information, the signal is broadly distributed and lacks strongly aligned directions. In contrast, SAEs exhibit a characteristic pattern: many inactive or low-specificity units and a distinct minority of high-specificity features that reliably localize to individual substructures. These findings reinforce the notion that sparsity disentangles molecular representations into units that more closely resemble functional-group detectors, enabling sharper mechanistic interpretation.

**Fragment Specificity.**    We benchmark SAE features against dense model activations, Principal Component Analysis (PCA) projections, and random orthogonal rotations. We compile a set of popular SMARTS patterns representing key functional groups, rings, and atom contexts. For each representation, we compute the maximum feature specificity for each fragment, measuring the "best detector" available in that basis.

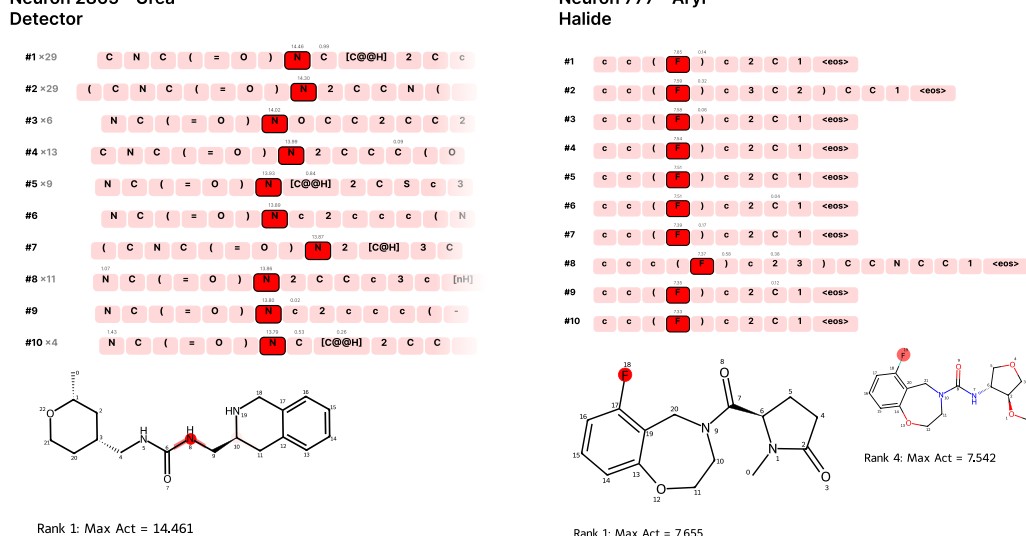

Figure 5: **Representative Chemical Features. (Left)** SAE feature with selective activation on the second nitrogen of urea groups (N–C(=O)–N). **(Right)** SAE feature that activates on fluorine atoms attached to aromatic carbons, with limited response to aliphatic fluorine, consistent with an aryl-halide detector.

As shown in Table 11 of the Appendix, SAE features consistently achieve higher interpretability scores compared to the raw residual stream and PCA components, despite PCA explaining more variance. This indicates that the SAE sparsity constraint successfully disentangles these concepts into distinct, high-specificity features that align with human-understandable chemical structures, while pure dimensionality reduction does not preserve integrity.

### 4.3 REPRESENTATIVE FEATURES

To illustrate the kind of structure recovered by SAEs, Figure 5 shows two representative latents discovered by the fragment-screening process. Additional qualitative examples can be found in Figure 9 in the appendix.

One feature activates almost exclusively on the second nitrogen in urea groups (N–C(=O)–N), with peak activation when the full motif has been written, suggesting a detector for completed urea linkers. Another feature activates on fluorine atoms attached to aromatic carbons, with much weaker response on aliphatic fluorine, consistent with an aryl-halide–like pattern. These examples indicate that individual SAE neurons can align with familiar medicinal chemistry motifs, although a broader, systematic survey is needed to assess how common such interpretable features are.

## 5 APPLICATIONS

### 5.1 DOWNSTREAM TASKS

**Activity Cliffs.** To assess whether the sparse autoencoders can encode pharmacologically relevant features, we evaluate their performance on MoleculeACE (van Tilborg et al., 2022). The chosen dataset focuses on bioactivity prediction and the identification of activity cliffs, i.e., pairs of structurally similar molecules with large differences in potency, constituting a challenging evaluation setting for machine learning architectures.

We evaluate SAE-based features against multiple baseline types. These include (1) raw dense transformer embeddings underlying the sparse activations, (2) standard Extended Connectivity Fingerprints (ECFP), and (3) supervised deep learning architectures reported in the benchmark's repository.

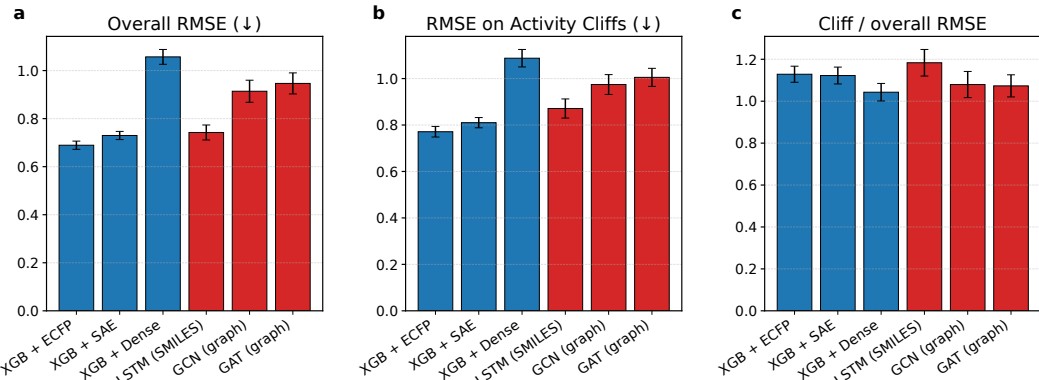

Figure 6: **Performance on MoleculeACE.** Comparison of mean performance across 30 endpoints – lower is better. **(a)** Overall RMSE on all compounds. **(b)** RMSE on activity cliffs only. **(c)** Ratio of cliff RMSE to overall RMSE (lower values indicate relatively better performance on cliffs). Bars show means across datasets and seeds; error bars denote 95% confidence intervals. Blue bars correspond to XGBoost models with different feature sets (ECFP, SAE, dense transformer embeddings), red bars to supervised deep learning baselines from van Tilborg et al. (2022).

As shown in Figure 6, SAE features significantly outperform dense transformer embeddings, achieving a mean RMSE of 0.730 across 30 endpoints, compared to 1.057 for the raw dense embeddings. Notably, unsupervised SAE features outperform established supervised deep learning baselines, including LSTMs (Hochreiter & Schmidhuber, 1997) and Graph Convolutional Networks (Kipf & Welling, 2017). While the ECFP fingerprint baseline remains the strongest predictor overall (0.689 RMSE) in line with the original evaluation from van Tilborg et al. (2022), the SAE features offer a competitive data-driven alternative that does not rely on hard-coded chemical invariants. More details are found in Table 12 and Table 13 of the appendix.

**ADME Tasks.** We also evaluate sparse features on four cADME assays from Fang et al. (2023): human liver microsomal clearance (HLM), P-gp efflux (MDR1), rat liver microsomal clearance (RLM), and kinetic solubility to test whether SAEs are able to capture pharmacokinetically relevant structures. As shown in Table 2, supervised graph models (MPNN1, MPNN2) achieve the highest Pearson correlations overall ($r \approx 0.62$–$0.74$). Unsupervised SAE features outperform both classical ECFP fingerprints and the FCFP4+rdMD descriptor baseline used in the original study, with consistent gains across endpoints (e.g., HLM: 0.617 vs. 0.578, RLM: 0.659 vs. 0.605). On MDR1, SAE features even slightly surpass the weaker graph baseline (MPNN1).

Our findings confirm that disentangling transformer representations improves performance on practical tasks, and also show that self-supervised features from a generative transformer can narrow the gap to more specialized models without carefully engineered chemical priors. Additional results are found in Section E.4.

| Task | Baselines | | Graph Models | | Ours |
|------|-----------|-------------|--------|--------|------|
| | **ECFP** | **FCFP4 + rdMD** | **MPNN1** | **MPNN2** | **SAE** |
| HLM | 0.578 | 0.575 | 0.676 | 0.714 | 0.617 |
| MDR1 | 0.656 | 0.643 | 0.666 | 0.735 | 0.676 |
| RLM | 0.605 | 0.575 | 0.727 | 0.735 | 0.659 |
| Solubility | 0.461 | 0.502 | 0.618 | 0.638 | 0.513 |

Table 2: **Performance on cADME Benchmarks.** Mean Pearson correlation coefficient ($r$) across 10 random scaffold-split seeds. FCFP4+rdMD refers to the method established in Fang et al. (2023) using FCFP4 fingerprints and RDKit descriptors. While fully supervised models consistently have the highest performance, unsupervised SAE features outperform both ECFP fingerprints and the manually engineered descriptor baseline used in the original study.

## 5.2 FEATURE STEERING

We share preliminary results on whether SAE-derived features can be used to steer molecular generation at inference time without retraining. For a panel of drug-relevant reference molecules, we extract max-pooled SAE latents and inject a scaled copy of this vector into the residual stream during sampling. Across most targets and steering settings, this procedure increases pairwise Tanimoto similarity among generated samples, concentrating exploration around a more coherent region of chemical space. For a subset of targets we also observe a clear improvement in Tanimoto similarity between individual samples and the conditioning molecule, indicating that latent steering can bias the generator toward user-specified structures. Example visualizations can be found in Figure 7, and Figure 10 in Appendix G compares distributions of steered and baseline examples.

These findings on SAE-based latent steering suggest that sparse features can provide an additional handle for controllable exploration using only a small number of activations. We note that the effect is not uniform. The steering coefficient and layer choice matter, and for more complex targets or aggressive steering strengths, validity may decrease drastically. We also point out this kind of steering does not constitute a precise "free lunch" conditional generator, but should rather be treated as a tool for high-throughput SMILES generation to focus samples around a desired region of molecular space.

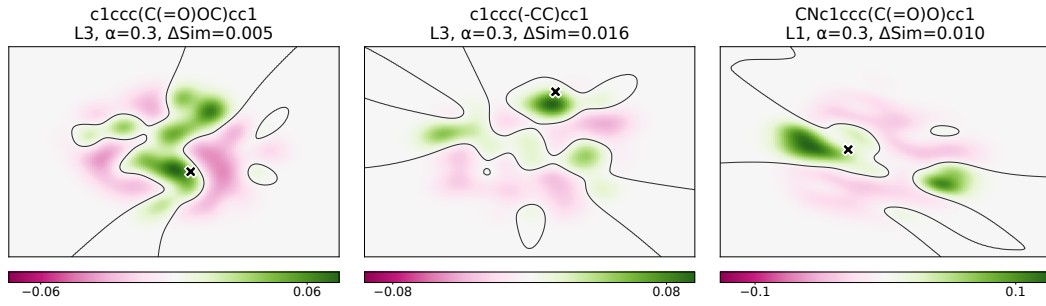

Figure 7: **SAE Steering Reshapes Exploration.** Density-difference UMAPs for three reference molecules, comparing baseline samples to SAE-steered generations by distance to target. Steering frequently shifts sampling toward a compact neighborhood around the target while preserving coverage of nearby, chemically related regions. Cross denotes target molecule.

## 6 CONCLUSION

We have presented a mechanistic analysis of autoregressive transformers trained on drug-like molecules, providing evidence for computational patterns that can explain aspects of chemistry-aligned generation in these models. We identified specialized attention heads that implement pointer algorithms for syntactic parsing, including multi-head circuits for ring closures and dedicated mechanisms for branch balancing. Our investigation revealed that fundamental concepts like valence capacity are encoded as distributed linear features in the residual stream, finding a single causal direction that monotonically controls bond-order predictions. Using sparse autoencoders, we presented an approach to shortlist interpretable features for manual elicitation, enabling faster identification of pharmacologically relevant representations.

The mechanistic patterns we identify provide concrete, testable handles on how this model organizes chemistry-aligned information, but we regard them primarily as descriptive. Our analyses of circuits, valence-aligned directions, and sparse features suggest potential routes toward more targeted interventions, more fine-grained failure characterization, and more informed model-design choices, but we leave a systematic exploration of these applications to future work.

While our analysis focuses on a specific model architecture and molecular representation, the methodological framework we present is broadly applicable to understanding how language models encode domain-specific knowledge. This work suggests that mechanistic language model interpretability techniques can provide insights into molecular machine learning systems, though significant methodological challenges remain for further investigation.

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

## A RELATED AND FUTURE WORK

### A.1 RELATED WORK

Mechanistic interpretability tools similar to those we use here have recently been applied to biological sequence models, particularly protein language models, providing a close analogue of our approach in a different biochemical domain. Adams et al. (2025) train sparse autoencoders on the residual stream of ESM-2 and systematically evaluate how choices such as expansion factor, sparsity, and layer selection influence the emergence of biologically meaningful, monosemantic features, positioning SAEs as a bridge from mechanistic interpretability to mechanistic biology. Complementary work on InterPLM (Simon & Zou, 2025) uses SAEs and related transcoders to extract thousands of interpretable features, ties them to curated protein annotations, and releases tooling for interactive exploration, highlighting how sparse feature dictionaries can support large-scale scientific analysis. In Brixi et al. (2025), authors apply sparse autoencoders to the Evo 2 genome model and find that individual features align with phage insertions, operon structure, exon–intron boundaries, and protein secondary structures. Beyond interpretation, Villegas Garcia & Ansuini (2025) show that SAE latents can be used to steer protein sequence generation toward specific structural motifs, treating sparse features as controllable handles in sequence design.

**Interpretability for Chemistry and Biology.** In the domains of chemistry and therapeutics, most interpretability studies focus on explaining predictions rather than characterizing the internal mechanisms of deep learning architectures. Examples include subgraph/feature attribution for GNNs (Pope et al., 2019; Ying et al., 2019) and model-agnostic counterfactuals that propose chemically coherent edits (Wellawatte et al., 2022). Broader reviews emphasize why transparent models matter in high-stakes discovery pipelines (Jiménez-Luna et al., 2020) and highlight successes such as AI-assisted antibiotic discovery (Stokes et al., 2020). Closer to our setting, Bagal et al. (2022) analyze attention-based saliency in an autoregressive SMILES transformer, and report heads that respond to branch parentheses, ring indices, and valence-related patterns. Their analysis demonstrates that syntactic and simple chemical regularities are reflected in attention maps. Our work complements these efforts by providing a more mechanistic treatment of molecular transformers.

**Broader Mechanistic Interpretability.** Early interpretability work on attention showed that many attention heads can be ablated or pruned with small loss in task performance (Michel et al., 2019), and sparked debate about whether attention weights themselves constitute faithful explanations (Jain & Wallace, 2019; Wiegreffe & Pinter, 2019). In contrast, mechanistic interpretability aims to reverse-engineer model internals as algorithmic *circuits*. The Transformer Circuits program formalized how attention heads compose to implement algorithms (Elhage et al., 2021) and connected

these ideas to concrete mechanisms such as induction heads for in-context learning (Olsson et al., 2022). Representation-level analyses via probing further characterized where linguistic information resides in pretrained encoders (Tenney et al., 2019a;b). Our syntactic findings for SMILES (pointer-style ring/branch circuits) draw inspiration from these works, but differ in both modality and objective: we analyze autoregressive *molecular* language models and show circuits specialized for discrete formal grammar constraints (ring digits, parentheses) and for chemically meaningful control signals (valence budgeting).

**Sparse Autoencoders.** SAEs have recently emerged as a practical tool to disentangle polysemantic activations and surface interpretable features in language models (Cunningham et al., 2023; Bricken et al., 2023). Subsequent variants improve fidelity or control sparsity, including TopK SAEs (Gao et al., 2024), Gated SAEs (Rajamanoharan et al., 2024a), and JumpReLU SAEs (Rajamanoharan et al., 2024b). We build on the baseline ReLU SAE setup and show that even this simple choice yields feature dictionaries that align with functional groups (e.g., urea, aryl-F) and provide competitive downstream signals in chemistry. Our fragment screening pipeline and universality experiments connect these representation-level tools to concrete chemical motifs and to stability across dictionary sizes.

**Molecular Language Models.** Transformers have been adapted to SMILES and related string representations for both generation and property prediction. Representative encoder models include ChemBERTa (Chithrananda et al., 2020) and MolBERT (Fabian et al., 2020); larger-scale encoder LMs such as MolFormer (Ross et al., 2022) demonstrate the benefits of rotary embeddings and efficient attention at scale. Autoregressive decoders such as MolGPT (Bagal et al., 2022) show that transformer language models can generate valid, diverse molecules and already provide qualitative evidence that attention heads track SMILES syntax (e.g., ring labels and branch structure). Our analysis targets a similar class of autoregressive models, but moves from saliency-style inspection to local causal interventions and sparse feature extraction, providing a more granular account of how syntax circuits and chemically meaningful residual features are implemented. We further connect these internal mechanisms to downstream benchmarks, suggesting that the same structures that support valid SMILES generation also give rise to useful pharmacological signals.

## A.2 FUTURE WORK

Our results point to several directions that may deepen mechanistic understanding of molecular transformers and strengthen their practical use in molecular design.

**Model-Internal Control Signals.** The valence-aligned direction and sparse feature activations identified in our analysis suggest that autoregressive transformers contain internal signals that can modulate structural decisions during generation. A natural next step is to determine how stable these signals are across training regimes, and whether architectural or regularization strategies can encourage more robust or disentangled control channels. Such work may clarify how to expose controllable degrees of freedom in SMILES-based generators without sacrificing validity or diversity.

**Beyond a Single Model and Representation.** Our study focuses on a single decoder-only architecture trained on canonical SMILES. Extending this analysis to masked models, encoder–decoder systems, and graph-augmented variants would help determine which mechanisms—such as pointer-style syntax circuits or valence features—are universal to molecular language models. Similarly, comparing models trained with randomized SMILES (Arús-Pous et al., 2019; Bjerrum, 2017) could reveal which behaviors reflect chemical structure rather than representational artifacts.

**Mechanisms Under Task Conditioning.** We investigate mechanisms for unconditional generation, but many practical settings involve explicit objectives (e.g., bioactivity or physicochemical targets). Studying how the identified circuits and residual features adapt under task conditioning may illuminate how supervised signals shape internal structure, and whether new interpretable directions emerge when the model is required to satisfy domain constraints.

**Improving Sparse Feature Dictionaries.** Sparse autoencoders recover many chemically interpretable features, but deeper-layer latents remain specialized and sometimes difficult to annotate.

Exploring alternative sparse autoencoder architectures, such as gated or thresholded SAEs (Gao et al., 2024; Rajamanoharan et al., 2024a;b), adopting transcoder architectures (Dunefsky et al., 2024; Ameisen et al., 2025) as well as other feasible dictionary learning methods, and examining how feature sets evolve across scales may enhance the coherence and identifiability of learned dictionaries. Understanding when features split or refine as capacity increases could clarify which representations correspond to stable chemical primitives.

**Integration with Downstream Workflows.** Our downstream experiments indicate that sparse features capture useful pharmacological signals complementary to fingerprints and GNN-based embeddings. Future work could integrate mechanistic insights more directly into design loops, including active learning, virtual screening, and controllable generation. Interpretable failure modes—such as disrupted valence tracking or missing substructure detectors—may also guide principled augmentation strategies or small architectural changes that improve robustness.

Overall, these directions aim to translate descriptive analysis into a clearer picture of how chemical structure is represented inside molecular transformers, and how these insights may inform both model design and more reliable molecular generation.

## B  SCOPE AND LIMITATIONS

**Syntax Circuits.** The pointer head analysis focuses on syntactic correctness in SMILES generation, where parentheses and ring closures represent critical structural constraints. While this approach is well-suited to the model's training objective and the computational challenges of long-range dependency matching, our findings are limited to correlational and ablation-based evidence. The observed attention patterns and causal effects demonstrate that these heads contribute to syntactic correctness, but do not rule out alternative or complementary computational pathways.

**Valence Budgeting.** This analysis focuses on explicit bond decisions in drug-like molecules, where standard organic valence rules apply and explicit bond symbols represent critical generation decisions. While this scope aligns with the model's lead-like training data and intended applications, these findings are necessarily limited to correlational evidence of mechanism. Linear probes may detect accessible statistical patterns rather than the model's actual computational pathway, and token-level analysis captures a local view of what may be broader processes. These constraints are important for interpreting our mechanistic claims, though they do not diminish the practical relevance of understanding how molecular transformers handle chemical constraints during generation.

## C  ETHICAL CONSIDERATIONS

This work presents methods for molecular transformers interpretability, which introduces several ethical considerations. While our focus is on interpretability of drug-like molecules for legitimate pharmaceutical applications and the primary goal of our study is to provide transparency in this domain, the mechanistic insights could theoretically inform efforts to generate harmful compounds or be applied to other chemical domains. We recommend that researchers and practitioners evaluate risk factors and implement appropriate safeguards, especially when working with sensitive data and large-scale drug discovery systems.

Our interpretability analysis reveals both capabilities and limitations of molecular transformers, emphasizing that these models should complement rather than replace human expertise in chemical design. The mechanistic insights we provide should complement experimental validation and traditional chemical knowledge rather than replace it. While the molecular transformers used in this work are relatively small-scale and computationally inexpensive compared to large language models, we encourage continued consideration of computational and energy requirements in scaling interpretability methods to larger chemical datasets.

## D  USE OF LARGE LANGUAGE MODELS

Large language models (LLMs) were used regularly throughout research to support various aspects of the work. While such systems certainly impacted aspects of this work, scientific conclusions,

structure and content of experiments, and interpretations represent the authors' independent judgment. The core contributions, including the mechanistic insights and interpretability methodology, are products of human-designed experiments and analysis. Specifically, commercial language models from various providers were employed for:

1. **Research**: LLMs were used as brainstorming partners to explore research directions, refine hypotheses, and suggest experimental designs for mechanistic interpretability approaches.

2. **Coding**: LLMs provided assistance in identifying and resolving bugs in analysis code, particularly for sparse autoencoder training, feature screening pipelines, and statistical analysis scripts. Language models helped refine and glue together experiments.

3. **Visuals**: LLMs helped generate LaTeX code for tables and figures, format results presentations, and create plots for experimental findings.

4. **Writing**: LLMs provided assistance in writing, editing, and structuring portions of this manuscript, including literature review, methodology descriptions, and results interpretation. All LLM-generated content was reviewed, fact-checked, and corrected by the authors.

## E ADDITIONAL RESULTS

### E.1 SAE REPRESENTATION QUALITY

To ensure that the learned sparse features are faithful representations of the model's internal state, we evaluate the SAEs by intervening during the forward pass. We replace the original residual-stream activations $x$ with the SAE reconstructions $\hat{x} = D(E(x))$ and measure the impact on both next-token prediction and generative outcomes. Table 10 summarizes these metrics across layers and expansion factors.

**Reconstruction Fidelity.** We measure fidelity using the increase in Cross Entropy ($\Delta$ CE) and the Kullback-Leibler Divergence ($D_{KL}$) between the model's output distribution using original versus reconstructed activations. Key observations are as follows.

- **Scaling:** Consistent with expectations from natural language, increasing the dictionary size from 4x to 16x generally improves reconstruction fidelity. In Layer 1, for example, the 16x SAE reduces $\Delta$ CE by 50% compared to the 4x baseline ($0.032 \rightarrow 0.016$).

- **Mid-network Bottleneck:** Reconstruction difficulty peaks in the middle layers (L3 and L4), where $\Delta$ CE reaches its maximum ($\sim 0.10$). This suggests that the model's representations are most information-dense or non-linear at this depth, requiring higher capacity to capture faithfully. The final layer (L5) is easily reconstructed ($\Delta$ CE $\approx 0.03$), likely because its representation is more closely aligned to the transformer's language modeling head.

**Sampling Capabilities.** We assess whether SAE-based interventions produce valid, high-quality molecules by generating new samples that use edited transformer activations that are decoded from sparse activations.

- **Chemical Validity:** Across all layers and sizes, the validity of generated molecules remains exceptionally high ($> 97\%$). In the final layer, reconstructed activations achieve $100\%$ validity. These results confirm both the triviality of last-layer residual stream activations, and that the sparse features capture a portion of syntactic constraints required to construct legal SMILES strings.

- **Sample Diversity:** We examine the diversity of generated scaffolds. 16x SAEs consistently preserve a higher number of unique scaffolds compared to smaller dictionaries (e.g., in Layer 1: 6041 scaffolds for 16x vs. 5908 for 4x). This result shows that larger dictionaries are better at capturing rare/long tail chemical features that contribute to overall structural diversity.

Table 10 summarizes key evaluation metrics for the sparse autoencoders trained on each transformer layer, including reconstruction and sparsity measures.

## E.2 FEATURE UNIVERSALITY

A known limitation of sparse dictionary learning is the potential for non-identifiability: recent work has shown that SAEs trained on the same data with different random seeds can converge to disjoint feature sets, suggesting that many discovered directions may be arbitrary rather than fundamental (Paulo & Belrose, 2025). Consequently, demonstrating that features persist across different training runs or hyperparameters is critical for establishing their mechanistic validity.

To address this question, we performed a cross-model comparison across dictionary sizes. For every feature in a smaller SAE, we identified its best match in larger SAEs trained on the same layer. We compute the maximum cosine similarity (MCS) between decoder weight vectors:

$$\max_{v_{\text{large}} \in D_{\text{large}}} \left( \frac{v_{\text{small}} \cdot v_{\text{large}}}{\|v_{\text{small}}\| \|v_{\text{large}}\|} \right)$$

We also computed a baseline using random unit vectors to establish the floor of high-dimensional alignment. As shown in Table 3, the stability of learned features varies by layer depth:

- **Stable Early Recovery:** Layer 1 demonstrates the highest degree of cross-model alignment, with a mean MCS of 0.940 between the 8x and 16x autoencoders. Notably, over 80% of the features in the smaller dictionary are recovered with strict fidelity ($> 0.9$ cosine similarity) in the larger dictionary. This indicates that for early processing layers, the sparse features extracted are largely independent of the dictionary size, suggesting the model relies on a stable set of low-level representations.

- **Similarity-Recovery Divergence:** In deeper layers (L3–L5), we observe a divergence between mean similarity and strict recovery. While the mean MCS remains high ($\sim 0.90$), indicating that the feature subspaces are well-aligned, the rate of strict feature recovery drops to 55-65%.

High general alignment but lower one-to-one correspondence is consistent with behaviors observed in larger language models where features in smaller dictionaries map to multiple, more granular features in larger dictionaries, frequently referred to as feature splitting (Bricken et al., 2023), or where the representation becomes inherently more polysemantic, although verifying whether this phenomenon is prevalent in molecular transformers requires further elicitation and is left to future work by the authors.

## E.3 FRAGMENT FEATURE ANALYSIS

To further contextualize the differences between dense and sparse representations, we examined kernel-density estimates of fragment specificity (WSD) across several common chemical motifs. The resulting distributions, as previously shown in Figure 4 reveal a consistent structure across layers and dictionary sizes. Dense residual activations show broad, smoothly varying specificity profiles, reflecting the well-known polysemanticity of transformer features: fragment information is present, but intermingled with unrelated contextual signals.

In contrast, SAE dictionaries exhibit a bimodal pattern. The large spike at zero specificity corresponds to the sparsity prior, which encourages most features to remain inactive for a given chemical context. More importantly, the right-tail extends substantially further than in the dense basis, indicating the presence of a small number of highly selective features. These outlier neurons correspond to interpretable, functional-group–aligned detectors (e.g., aryl fluorine, carbonyl, urea), many of which recur across molecules with diverse scaffolds. These observations provide quantitative evidence that SAEs carve chemically coherent structure out of otherwise diffuse internal activations, supporting their use as a practical interpretability tool for molecular language models.

## E.4 TDC BENCHMARKS

To validate whether SAE features capture pharmacologically relevant information, we further evaluate their performance on a suite of popular downstream prediction tasks using Therapeutics Data Commons (TDC) (Huang et al., 2021) endpoints. The chosen tasks cover a diverse range of endpoints in drug discovery, including ADMET properties (e.g., BBB permeability, P-gp inhibition), safety pharmacology (hERG cardiotoxicity), and pharmacokinetics (Caco-2 permeability, Plasma Protein Binding). These results are summarized in Figure 8.

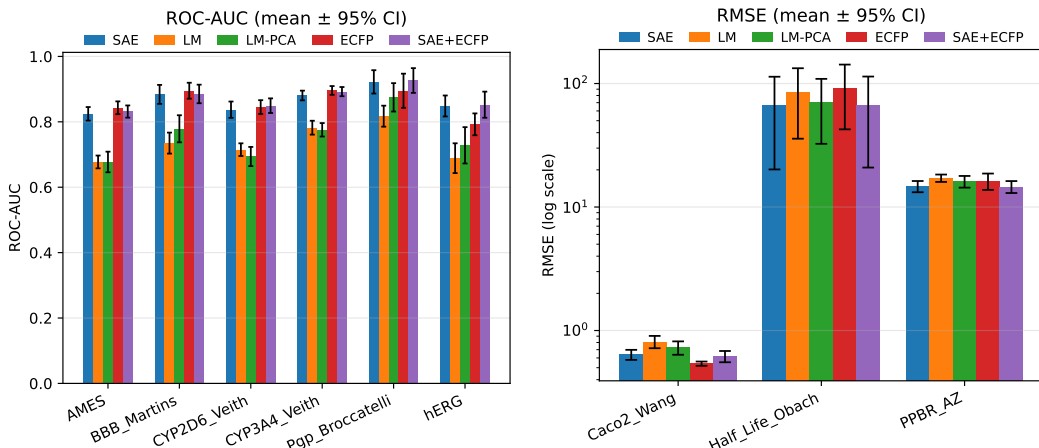

Figure 8: **Performance on Therapeutics Data Commons.** SAE features consistently outperform LM and LM-PCA embeddings and are competitive with, or complementary to, ECFP fingerprints across TDC endpoints. Bars show mean performance with 95% confidence intervals over scaffold-split seeds; regression metrics are plotted on a log scale.

SAE features are compared against the dense transformer embeddings from which they were derived, providing one measure of the value added by sparsification. We include a PCA baseline on the embeddings to test whether a simple linear dimensionality reduction can match the SAE's non-linear feature extraction. Finally, we benchmark against ECFP fingerprints, a widely used non-machine learning method in cheminformatics, to situate our performance against a respected domain-specific baseline. Finally, we test a simple concatenation (SAE+ECFP) to probe for complementarity between the learned and engineered features.

**Classification.** Across the six TDC classification tasks in Table 5, sparse autoencoder (SAE) features consistently outperform raw transformer (LM) and PCA baselines, and are competitive with or better than ECFP fingerprints. The largest gain appears on hERG cardiotoxicity (0.8485 vs. 0.7923 ROC-AUC), suggesting that SAE features capture complementary ion-channel–relevant structure. Combining SAE features with ECFP typically yields the strongest or second-strongest performance, indicating that learned sparse representations add information beyond standard fingerprints.

**Regression.** As shown in Table 7, for Plasma Protein Binding (PPBR), SAE-based features outperform all other baselines, with the combined SAE+ECFP representation achieving the lowest RMSE (14.60), suggesting that SAE features effectively encode properties that govern protein binding. Despite high variance across all methods on the half-life prediction task, SAE features achieve the lowest mean RMSE. These patterns are consistent with SAE features capturing distributed, context-dependent signals that may complement ECFP's fragment-local encodings.

**Comparison with Stronger Baselines.** To contextualize the predictive quality of the extracted features, we benchmarked them against popular baselines from the TDC leaderboard, including MapLight-GNN (Notwell & Wood, 2023) and Chemprop-RDKit (Yang et al., 2019). As shown in Table 6, our unsupervised SAE features are highly competitive with fully supervised GNNs, notably surpassing Chemprop on endpoints such as hERG cardiotoxicity (0.849 vs 0.840 ROC-AUC) and P-glycoprotein inhibition (0.922 vs 0.886 ROC-AUC). While architectures like MapLight-GNN remain superior on some tasks, the ability of sparse autoencoders to recover high-fidelity pharmacological signals purely from a self-supervised generation objective confirms that the model has internalized an adequate representation of molecular topology.

## F IMPLEMENTATION DETAILS

All experiments were implemented in JAX. Transformer and sparse autoencoder training, as well as code for running experiments described in this work, ran on a single NVIDIA Grace Hopper GH200

instance. In this section, we share further details about the specific architectures, hyperparameters, and datasets used throughout the paper's development.

### F.1 TRANSFORMER ARCHITECTURE

We trained a decoder-only transformer from scratch. The model consists of 6 layers with 8 attention heads, a hidden dimension of $d = 512$, and an intermediate feed-forward dimension of $d_{\text{ff}} = 2048$. Our code implements a LLaMA-style architecture (Touvron et al., 2023), incorporating RMSNorm ($\epsilon = 10^{-6}$) applied before attention and feed-forward blocks, and SiLU activations. Positional information was encoded using Rotary Positional Embeddings (RoPE) with a base frequency of $\theta = 100,000$. The model was trained with a context window of 256 tokens.

Training was performed using the Muon optimizer (Jordan et al., 2024), a momentum-orthogonal optimizer designed for efficient large-scale training. We used a peak learning rate of $5 \times 10^{-4}$ with a linear warmup of 3000 steps, followed by a cosine decay schedule to zero. The global batch size was set to 4096 sequences. Regularization included a weight decay of 0.01 and global gradient norm clipping at 1.0.

### F.2 SAE TRAINING

We trained separate SAEs on the residual stream of each transformer layer (0-5 for the 6-layer model), with three dictionary sizes: 4x (2048), 8x (4096), and 16x (8192). These models utilize a standard architecture with a few modifications to improve feature quality and prevent neuron death.

As described in Section 2.1, our SAEs consist of a linear encoder with a ReLU nonlinearity and a linear decoder with tied biases and a unit-norm constraint enforced on decoder columns to prevent scale ambiguity. The training objective minimizes the linear combination of reconstruction error (MSE) and sparsity ($L_1$).

All training runs used AdamW (Loshchilov & Hutter, 2019) with a cosine decay schedule, and learning rates were swept per layer to ensure stable convergence. To reduce the number of dead neurons, we added ghost grads (Jermyn & Templeton, 2024) using a ghost coefficient of $0.1$. We tuned sparsity $\lambda$ per layer to target an average $L_0$ loss between 10 and 50.

### F.3 DATASET COMPOSITION

We used the lead-like subset of ZINC20, consisting of approximately 383M small molecules initially filtered by ZINC criteria. To refine this dataset towards drug-like compounds, we implemented a filtering cascade using RDKit. First, salts were stripped, and molecules were retained only if they satisfied the following property constraints: molecular weight between 150 and 500 Da, calculated LogP (Crippen's LogP) between -1.0 and 5.0, $\leq 5$ hydrogen bond donors, $\leq 10$ hydrogen bond acceptors, topological polar surface area between 20 and 140 Å$^2$, $\leq 10$ rotatable bonds, and between 1 and 6 rings inclusive.

Molecules were filtered to remove known Pan-Assay Interference Compounds (PAINS) using RDKit's built-in filter. Finally, only molecules with a Quantitative Estimate of Drug-likeness (QED) score greater than 0.4 were kept. The canonical SMILES representation of the molecules passing all filters constituted our final pretraining dataset.

To ensure proper evaluation without scaffold leakage, we employed a scaffold-based splitting procedure. For each molecule, we extracted its Murcko scaffold using RDKit and assigned it to a deterministic bucket based on a hash of its canonical SMILES. This approach grouped structurally similar molecules together and prevented related structures from appearing across splits. We allocated 70% of scaffold buckets to the training set, 10% to validation, and 20% to test sets, resulting in approximately 244M training, 30M validation, and 60M test compounds.

### F.4 EXPERIMENTAL DETAILS

**TDC Downstream Evaluation.** For all datasets, we employ scaffold-based splitting (70/10/20 split for training, validation, and testing) repeated across three random seeds to test generalization to novel chemical structures. We compare four distinct feature representations: baseline Extended

Connectivity Fingerprints (ECFP, radius 2, 2048 bits), dense language model embeddings derived by mean-pooling the final residual stream, SAE-derived sparse features aggregated via max-pooling across the sequence, and a dimensionality-controlled baseline using PCA-reduced dense embeddings (256 dimensions). We utilize Gradient Boosted Trees (XGBoost) as the prediction head to evaluate informational content rather than linear separability, using hyperparameters of 2000 estimators, a maximum depth of 6, a learning rate of 0.05, and a subsample ratio of 0.8, with early stopping triggered after 50 rounds of no improvement on the validation set. Results are reported as the mean ROC-AUC for classification and RMSE for regression, with 95% confidence intervals calculated via Student's $t$-distribution.

**SAE Steering Analysis.** To quantify the extent to which SAE latents can be used for controllable generation, we performed a steering benchmark that treats the sparse representation of a reference molecule as a soft conditioning signal. We first curated a set of approximately 500 drug-relevant, relatively simple molecules (heterocycles, common functional groups, bioisosteres, and small drug fragments, etc.). For each target SMILES, we generate a baseline batch of molecules using the pretrained transformer without any interventions and compute (i) pairwise Tanimoto similarity among the samples and (ii) Tanimoto similarity between each sample and the target molecule, using 2048-bit Morgan fingerprints.

For steering, we use the trained sparse autoencoders to encode the target at a chosen layer $L$, obtaining a max-pooled latent vector $z_{\text{ref}} \in \mathbb{R}^m$. To focus the intervention on salient features, we keep only the top-$k$ activations ($k = 5$ in the shared experiments) and set all others to zero. During autoregressive sampling, we register an activation hook at the corresponding SAE placement and modify the residual stream via

$$z_{\text{new}} = z_{\text{cur}} + \alpha \, z_{\text{ref}}, \tag{2}$$

where $z_{\text{cur}}$ are the current SAE latents and $\alpha$ is a scalar steering coefficient. The modified latents are then decoded back to the residual stream using the SAE decoder, leaving the rest of the model unchanged. We sweep layers $L \in \{0, \ldots, 5\}$ and steering strengths $\alpha \in \{0.2, 0.4, 0.6, 0.8\}$, generating $N = 100$ samples per (target, layer, $\alpha$) configuration, and log both SMILES validity and similarity statistics. A fixed, larger unsteered batch is also generated once to provide a global baseline for validity and target similarity.

Across the majority of targets and steering settings, we observe a positive shift in pairwise Tanimoto similarity among steered samples relative to the baseline generator, indicating that the intervention tends to concentrate exploration into a more coherent local neighborhood of chemical space. For a non-trivial subset of targets, steering also increases the mean Tanimoto similarity to the conditioning molecule, sometimes substantially, suggesting that a share of the injected SAE features act as a meaningful bias toward the reference structure rather than merely collapsing diversity. However, the effect of $\alpha$ is not uniform: larger coefficients or steering at deeper layers can degrade validity, especially for longer or structurally complex targets, and in some settings the similarity gains vanish or reverse. Overall, these experiments should be viewed as preliminary evidence that SAE latents can be used as lightweight control knobs over generation, rather than as fully fledged conditional generators.

# G Supplementary Tables and Figures

| Layer | Comparison | MCS | Recovery | Baseline |
|---|---|---|---|---|
| 0 | 4x$\to$ 8x | 0.623 | 46.3% | 0.157 |
| | 4x$\to$ 16x | **0.625** | **46.6%** | 0.166 |
| | 8x$\to$ 16x | 0.406 | 22.7% | 0.166 |
| 1 | 4x$\to$ 8x | 0.938 | 79.3% | 0.151 |
| | 4x$\to$ 16x | 0.923 | 71.8% | 0.160 |
| | 8x$\to$ 16x | **0.940** | **82.0%** | 0.161 |
| 2 | 4x$\to$ 8x | 0.915 | 68.0% | 0.154 |
| | 4x$\to$ 16x | 0.880 | 49.2% | 0.159 |
| | 8x$\to$ 16x | **0.927** | **74.2%** | 0.159 |
| 3 | 4x$\to$ 8x | 0.895 | 58.0% | 0.157 |
| | 4x$\to$ 16x | 0.857 | 38.4% | 0.162 |
| | 8x$\to$ 16x | **0.910** | **65.2%** | 0.162 |
| 4 | 4x$\to$ 8x | 0.861 | 45.1% | 0.156 |
| | 4x$\to$ 16x | 0.818 | 27.3% | 0.164 |
| | 8x$\to$ 16x | **0.898** | **60.7%** | 0.164 |
| 5 | 4x$\to$ 8x | 0.792 | 35.1% | 0.151 |
| | 4x$\to$ 16x | 0.773 | 26.7% | 0.157 |
| | 8x$\to$ 16x | **0.862** | **52.8%** | 0.157 |

Table 3: **Feature Stability.** We measure how well features from a smaller SAE are recovered by a larger SAE by computing cosine similarity between activations of different autoencoders. Rightmost column includes random baseline performance for comparison. Layer 1 shows exceptional stability ($> 70\%$ strict recovery), indicating a robust basis for chemical primitives.

| Layer | Size | Jaccard Mean | Jaccard Std | Cosine Mean | Cosine Std |
|---|---|---|---|---|---|
| 0 | 4x | 0.935 | 0.013 | 0.798 | 0.345 |
| | 8x | 0.979 | 0.004 | 0.928 | 0.227 |
| | 16x | **0.993** | 0.002 | **0.976** | 0.134 |
| 1 | 4x | 0.677 | 0.032 | 0.406 | 0.376 |
| | 8x | 0.818 | 0.022 | 0.470 | 0.432 |
| | 16x | **0.939** | 0.009 | **0.707** | 0.420 |
| 2 | 4x | 0.461 | 0.034 | 0.304 | 0.256 |
| | 8x | 0.605 | 0.030 | 0.317 | 0.359 |
| | 16x | **0.844** | 0.018 | **0.493** | 0.450 |
| 3 | 4x | 0.511 | 0.035 | **0.392** | 0.248 |
| | 8x | 0.383 | 0.034 | 0.217 | 0.223 |
| | 16x | **0.579** | 0.029 | 0.255 | 0.343 |
| 4 | 4x | **0.606** | 0.031 | **0.507** | 0.269 |
| | 8x | 0.379 | 0.034 | 0.229 | 0.209 |
| | 16x | 0.340 | 0.031 | 0.158 | 0.205 |
| 5 | 4x | **0.595** | 0.030 | **0.471** | 0.282 |
| | 8x | 0.327 | 0.030 | 0.179 | 0.174 |
| | 16x | 0.220 | 0.032 | 0.088 | 0.102 |

Table 4: **Augmentation Robustness.** Table describes SAE feature robustness metrics by layer and expansion factor. Jaccard similarity measures token-level overlap, while cosine similarity measures feature activation vector alignment. Bold values indicate best performance within each layer group.

| Task | SAE | LM | PCA | ECFP | SAE+ECFP |
|---|---|---|---|---|---|
| **Classification (ROC-AUC)** | | | | | |
| AMES | $0.8245 \pm 0.0207$ | $0.6773 \pm 0.0196$ | $0.6772 \pm 0.0315$ | $\mathbf{0.8430 \pm 0.0196}$ | $\underline{0.8314 \pm 0.0187}$ |
| BBB_Martins | $0.8840 \pm 0.0290$ | $0.7349 \pm 0.0321$ | $0.7787 \pm 0.0414$ | $\mathbf{0.8952 \pm 0.0243}$ | $\underline{0.8852 \pm 0.0284}$ |
| CYP2D6_Veith | $0.8372 \pm 0.0249$ | $0.7149 \pm 0.0194$ | $0.6939 \pm 0.0293$ | $\underline{0.8451 \pm 0.0211}$ | $\mathbf{0.8494 \pm 0.0224}$ |
| CYP3A4_Veith | $0.8805 \pm 0.0148$ | $0.7821 \pm 0.0212$ | $0.7756 \pm 0.0206$ | $\mathbf{0.8961 \pm 0.0134}$ | $\underline{0.8924 \pm 0.0141}$ |
| Pgp_Broccatelli | $\underline{0.9222 \pm 0.0357}$ | $0.8173 \pm 0.0321$ | $0.8749 \pm 0.0434$ | $0.8950 \pm 0.0522$ | $\mathbf{0.9264 \pm 0.0378}$ |
| hERG | $\underline{0.8485 \pm 0.0321}$ | $0.6888 \pm 0.0456$ | $0.7282 \pm 0.0555$ | $0.7923 \pm 0.0334$ | $\mathbf{0.8524 \pm 0.0399}$ |
| **Classification (AU-PRC)** | | | | | |
| AMES | $0.8809 \pm 0.0099$ | $0.7638 \pm 0.0124$ | $0.7698 \pm 0.0097$ | $\mathbf{0.8888 \pm 0.0138}$ | $\underline{0.8843 \pm 0.0081}$ |
| BBB_Martins | $0.9539 \pm 0.0154$ | $0.8773 \pm 0.0235$ | $0.9027 \pm 0.0251$ | $\mathbf{0.9587 \pm 0.0110}$ | $\underline{0.9540 \pm 0.0141}$ |
| CYP2D6_Veith | $0.6134 \pm 0.0563$ | $0.3912 \pm 0.0502$ | $0.3481 \pm 0.0412$ | $\underline{0.6379 \pm 0.0622}$ | $\mathbf{0.6395 \pm 0.0595}$ |
| CYP3A4_Veith | $0.8431 \pm 0.0219$ | $0.7049 \pm 0.0122$ | $0.6943 \pm 0.0119$ | $\mathbf{0.8653 \pm 0.0179}$ | $\underline{0.8593 \pm 0.0215}$ |
| Pgp_Broccatelli | $\underline{0.9421 \pm 0.0375}$ | $0.8199 \pm 0.0733$ | $0.8840 \pm 0.0670$ | $0.9162 \pm 0.0633$ | $\mathbf{0.9458 \pm 0.0394}$ |
| hERG | $\underline{0.9074 \pm 0.0424}$ | $0.8016 \pm 0.0539$ | $0.8141 \pm 0.0878$ | $0.8720 \pm 0.0150$ | $\mathbf{0.9096 \pm 0.0409}$ |

Table 5: **Classification Tasks.** Classification results (ROC-AUC and AU-PRC) on TDC ADMET benchmarks for SAE features and other baselines. Values are mean $\pm$ 95% CIs across scaffold-split seeds. Best per row in bold; second best underlined. SAE+ECFP denotes concatenation.

| Task | Metric | SAE (Ours) | MapLight+GNN (TDC) | Chemprop-RDKit (TDC) |
|---|---|---|---|---|
| **Classification (ROC-AUC)** | | | | |
| AMES | ROC-AUC | $0.8245 \pm 0.0207$ | $\mathbf{0.869 \pm 0.002}$ | $\underline{0.850 \pm 0.004}$ |
| BBB_Martins | ROC-AUC | $\underline{0.8840 \pm 0.0290}$ | $\mathbf{0.913 \pm 0.001}$ | $0.869 \pm 0.027$ |
| Pgp_Broccatelli | ROC-AUC | $\underline{0.9222 \pm 0.0357}$ | $\mathbf{0.938 \pm 0.002}$ | $0.886 \pm 0.016$ |
| hERG | ROC-AUC | $\underline{0.8485 \pm 0.0321}$ | $\mathbf{0.880 \pm 0.002}$ | $0.840 \pm 0.007$ |
| **Classification (AU-PRC)** | | | | |
| CYP2D6_Veith | AU-PRC | $0.6134 \pm 0.0563$ | $\mathbf{0.790 \pm 0.001}$ | $\underline{0.673 \pm 0.007}$ |
| CYP3A4_Veith | AU-PRC | $0.8431 \pm 0.0219$ | $\mathbf{0.916 \pm 0.000}$ | $\underline{0.876 \pm 0.003}$ |

Table 6: **SAEs vs. Popular Baselines.** Comparison of SAE-based predictors with TDC leaderboard baselines on ADMET benchmarks. For each dataset, we report the metric used on the TDC leaderboard (ROC-AUC or AU-PRC) under scaffold split. Best per row in bold; second best underlined. MapLight+GNN and Chemprop-RDKit results are adopted from the public TDC Leaderboard at https://tdcommons.ai/benchmark/admet_group/overview/. Accessed on 23rd November, 2025.

| Task | SAE | LM | PCA | ECFP | SAE+ECFP |
|---|---|---|---|---|---|
| Caco2_Wang | 0.6379 | 0.8117 | 0.7266 | **0.5397** | 0.6181 |
| | ($\pm 0.0597$) | ($\pm 0.0927$) | ($\pm 0.0901$) | ($\pm 0.0212$) | ($\pm 0.0647$) |
| Half_Life_Obach | **66.8462** | 84.3876 | 70.7980 | 92.4908 | 67.3997 |
| | ($\pm 46.6882$) | ($\pm 48.6234$) | ($\pm 38.2962$) | ($\pm 49.9250$) | ($\pm 46.5255$) |
| PPBR_AZ | 14.7082 | 17.1500 | 16.1070 | 16.2194 | **14.6042** |
| | ($\pm 1.5350$) | ($\pm 1.1762$) | ($\pm 1.7440$) | ($\pm 2.4505$) | ($\pm 1.6209$) |

Table 7: **Regression Tasks.** RMSE values on TDC ADMET regression benchmarks comparing SAE input features with baselines. Values are mean $\pm$ 95% CIs across scaffold-split seeds. Best per row in bold; second best underlined.

| Layer | Acc. (%) | 95% CI |
|---|---|---|
| 0 | 81.36 | [80.78, 81.78] |
| 1 | 97.17 | [97.08, 97.37] |
| 2 | 98.96 | [98.78, 99.07] |
| 3 | 99.08 | [99.03, 99.14] |
| 4 | 98.85 | [98.74, 99.02] |
| 5 | 97.38 | [97.28, 97.51] |

| $\alpha$ | $\Delta$ single ($-$) | $\Delta$ double ($=$) | $\Delta$ triple (#) |
|---|---|---|---|
| -2.0 | +0.033 [0.029, 0.036] | -0.019 [-0.024, -0.015] | -0.053 [-0.056, -0.049] |
| -1.0 | +0.014 [0.013, 0.016] | -0.008 [-0.011, -0.006] | -0.027 [-0.029, -0.026] |
| -0.5 | +0.008 [0.007, 0.009] | 0.000 [-0.001, 0.001] | -0.011 [-0.012, -0.010] |
| 0.5 | -0.005 [-0.006, -0.004] | +0.002 [0.001, 0.004] | +0.008 [0.007, 0.009] |
| 1.0 | -0.009 [-0.010, -0.007] | +0.009 [0.007, 0.011] | +0.022 [0.020, 0.023] |
| 2.0 | -0.007 [-0.010, -0.004] | +0.017 [0.014, 0.021] | +0.047 [0.044, 0.051] |

Table 8: **Layer-wise Valence Probe Accuracy.** Accuracy and 95% CIs for valence prediction.

Table 9: **Effect of Valence Steering on Bond Logits.** Mean change in logits for bond tokens when intervening on Layer 3.

| Layer | Size | $\Delta$ CE | $D_{KL}$ | Validity | QED | $\Delta$ SA | Tanimoto | Scaffolds | Scaf. Overlap |
|---|---|---|---|---|---|---|---|---|---|
| 0 | 4x | **0.031** | **0.035** | **0.997** | 0.827 | -0.056 | **0.626** | **5895** | **0.150** |
| | 8x | 0.046 | 0.050 | 0.994 | **0.831** | -0.070 | 0.618 | 5817 | 0.147 |
| | 16x | 0.042 | 0.046 | 0.994 | 0.827 | **-0.073** | 0.615 | 5760 | 0.146 |
| 1 | 4x | 0.032 | 0.039 | 0.994 | **0.823** | -0.068 | 0.617 | 5908 | 0.143 |
| | 8x | 0.023 | 0.029 | 0.995 | 0.822 | -0.046 | 0.620 | 5980 | 0.147 |
| | 16x | **0.016** | **0.022** | **0.996** | 0.823 | -0.038 | **0.634** | **6041** | **0.158** |
| 2 | 4x | 0.063 | 0.073 | 0.977 | 0.821 | **-0.067** | 0.606 | 5643 | 0.127 |
| | 8x | 0.047 | 0.056 | 0.983 | **0.822** | -0.065 | 0.614 | 5840 | 0.132 |
| | 16x | **0.029** | **0.038** | **0.990** | 0.820 | -0.045 | **0.619** | **6019** | **0.142** |
| 3 | 4x | 0.103 | 0.116 | 0.979 | **0.833** | **-0.138** | 0.602 | 5455 | 0.115 |
| | 8x | 0.078 | 0.090 | 0.980 | 0.830 | -0.097 | 0.611 | 5771 | 0.122 |
| | 16x | **0.058** | **0.070** | **0.988** | 0.826 | -0.081 | **0.615** | **5900** | **0.135** |
| 4 | 4x | 0.105 | 0.117 | 0.993 | **0.831** | **-0.204** | 0.605 | 5063 | 0.119 |
| | 8x | 0.097 | 0.109 | 0.995 | 0.830 | -0.184 | 0.607 | 5273 | 0.124 |
| | 16x | **0.072** | **0.082** | **0.996** | 0.828 | -0.155 | **0.616** | **5428** | **0.130** |
| 5 | 4x | 0.033 | **0.038** | **1.000** | 0.822 | -0.089 | 0.609 | **5403** | 0.127 |
| | 8x | 0.032 | 0.039 | **1.000** | 0.820 | -0.118 | 0.612 | 5309 | 0.130 |
| | 16x | **0.031** | 0.040 | **1.000** | 0.821 | **-0.129** | 0.616 | 5312 | **0.132** |

Table 10: **Activation Replacements.** Evaluation metrics for SAE residual-stream activation editing by layer and expansion factor. All values are means across experiments.

| Fragment | Dense | Random | PCA (L0) | PCA (Full) | SAE (Ours) |
|---|---|---|---|---|---|
| Isopropyl | 0.00 ± 0.00 | **0.49 ± 0.00** | 0.49 ± 0.00 | 0.49 ± 0.00 | 0.20 ± 0.00 |
| Amide | 0.00 ± 0.00 | **0.49 ± 0.00** | 0.49 ± 0.00 | 0.49 ± 0.00 | 0.22 ± 0.00 |
| Chloro-Aromatic | 0.00 ± 0.00 | 0.02 ± 0.00 | 0.02 ± 0.00 | 0.02 ± 0.00 | **0.46 ± 0.03** |
| Ketone | 0.00 ± 0.00 | **0.44 ± 0.00** | 0.44 ± 0.00 | 0.44 ± 0.00 | 0.18 ± 0.00 |
| Nitrile | 0.00 ± 0.00 | 0.01 ± 0.00 | 0.01 ± 0.00 | 0.02 ± 0.00 | **0.44 ± 0.05** |
| Fluoro-Aliphatic | 0.00 ± 0.00 | 0.06 ± 0.00 | 0.06 ± 0.00 | 0.09 ± 0.00 | **0.42 ± 0.01** |
| Fluoro-Aromatic | 0.00 ± 0.00 | 0.03 ± 0.00 | 0.03 ± 0.00 | 0.03 ± 0.00 | **0.42 ± 0.01** |
| Thioether | 0.00 ± 0.00 | 0.09 ± 0.00 | 0.09 ± 0.00 | 0.09 ± 0.00 | **0.35 ± 0.01** |
| Sulfonyl | 0.00 ± 0.00 | 0.16 ± 0.00 | 0.16 ± 0.00 | 0.16 ± 0.00 | **0.35 ± 0.00** |
| Trifluoromethyl | 0.00 ± 0.00 | 0.05 ± 0.00 | 0.05 ± 0.00 | 0.07 ± 0.00 | **0.35 ± 0.01** |
| Carboxylic Acid | 0.00 ± 0.00 | 0.11 ± 0.00 | 0.11 ± 0.00 | 0.11 ± 0.00 | **0.32 ± 0.01** |
| Tert-Butyl | 0.00 ± 0.00 | **0.32 ± 0.00** | 0.32 ± 0.00 | 0.32 ± 0.00 | 0.18 ± 0.00 |
| Sulfonamide | 0.00 ± 0.00 | 0.14 ± 0.00 | 0.14 ± 0.00 | 0.14 ± 0.00 | **0.29 ± 0.01** |
| Alkene | 0.00 ± 0.00 | 0.05 ± 0.00 | 0.05 ± 0.00 | 0.06 ± 0.00 | **0.29 ± 0.01** |
| Nitro | 0.00 ± 0.00 | 0.01 ± 0.00 | 0.01 ± 0.00 | 0.02 ± 0.00 | **0.28 ± 0.05** |
| Ester | 0.00 ± 0.00 | 0.16 ± 0.00 | 0.16 ± 0.00 | 0.16 ± 0.00 | **0.27 ± 0.00** |
| Chloro-Aliphatic | 0.00 ± 0.00 | 0.01 ± 0.00 | 0.01 ± 0.00 | 0.02 ± 0.00 | **0.26 ± 0.03** |
| Ether | 0.00 ± 0.00 | 0.21 ± 0.00 | 0.21 ± 0.00 | 0.21 ± 0.00 | **0.26 ± 0.00** |
| Benzene | 0.00 ± 0.00 | 0.11 ± 0.00 | 0.13 ± 0.00 | 0.13 ± 0.00 | **0.24 ± 0.01** |
| Morpholine | 0.00 ± 0.00 | 0.15 ± 0.00 | 0.15 ± 0.00 | 0.15 ± 0.00 | **0.23 ± 0.00** |
| Urea | 0.00 ± 0.00 | 0.14 ± 0.00 | 0.14 ± 0.00 | 0.14 ± 0.00 | **0.23 ± 0.00** |
| Pyridine | 0.00 ± 0.00 | 0.13 ± 0.00 | 0.20 ± 0.00 | 0.20 ± 0.00 | **0.21 ± 0.00** |
| Piperidine | 0.00 ± 0.00 | **0.21 ± 0.00** | 0.21 ± 0.00 | 0.21 ± 0.00 | 0.18 ± 0.00 |
| Pyrimidine | 0.00 ± 0.00 | 0.14 ± 0.00 | **0.20 ± 0.00** | 0.20 ± 0.00 | 0.17 ± 0.00 |
| Phenol | 0.00 ± 0.00 | 0.05 ± 0.00 | 0.05 ± 0.00 | 0.05 ± 0.00 | **0.20 ± 0.01** |
| Imidazole | 0.00 ± 0.00 | 0.05 ± 0.00 | 0.07 ± 0.00 | 0.07 ± 0.00 | **0.18 ± 0.01** |
| Furan | 0.00 ± 0.00 | 0.04 ± 0.00 | 0.07 ± 0.00 | 0.07 ± 0.00 | **0.17 ± 0.02** |
| Cyclohexane | 0.00 ± 0.00 | 0.08 ± 0.00 | 0.08 ± 0.00 | 0.08 ± 0.00 | **0.17 ± 0.01** |
| Aniline | 0.00 ± 0.00 | 0.04 ± 0.00 | 0.04 ± 0.00 | 0.04 ± 0.00 | **0.15 ± 0.01** |
| Thiophene | 0.00 ± 0.00 | 0.02 ± 0.00 | 0.03 ± 0.00 | 0.03 ± 0.00 | **0.13 ± 0.01** |
| Aldehyde | 0.00 ± 0.00 | 0.01 ± 0.00 | 0.01 ± 0.00 | 0.01 ± 0.00 | **0.07 ± 0.02** |

Table 11: **Fragment Interpretability.** Fragment specificity (maximum WSD) across SMARTS patterns using various L3 residual-stream representations. Values are Mean ± 95% CI, calculated on 50K molecules.

Table 12: **MoleculeACE Performance**. Lower is better for RMSE / RMSE$_{cliff}$; a smaller ratio indicates relatively better cliff performance. Bold rows indicate comparisons with the same prediction head configuration. Values are means aggregated across benchmark endpoints.

| Source | Method | RMSE | RMSE$_{cliff}$ | RMSE ratio |
|---|---|---|---|---|
| MoleculeACE | SVM + ECFP | 0.671 | 0.751 | 1.127 |
| **Ours** | **XGB + ECFP Fingerprints** | **0.689** | **0.771** | **1.129** |
| MoleculeACE | GBM + ECFP | 0.695 | 0.783 | 1.138 |
| MoleculeACE | RF + ECFP | 0.700 | 0.786 | 1.137 |
| Ours | SVR + ECFP Fingerprints | 0.710 | 0.799 | 1.137 |
| **Ours** | **XGB + SAE Features** | **0.730** | **0.810** | **1.123** |
| MoleculeACE | LSTM (SMILES) | 0.742 | 0.871 | 1.183 |
| MoleculeACE | Transformer (tokens) | 0.896 | 0.970 | 1.091 |
| MoleculeACE | GCN (graph) | 0.914 | 0.974 | 1.080 |
| MoleculeACE | CNN (SMILES) | 0.930 | 0.968 | 1.051 |
| MoleculeACE | GAT (graph) | 0.947 | 1.005 | 1.073 |
| **Ours** | **XGB + Transformer Embeddings** | **1.057** | **1.088** | **1.043** |
| MoleculeACE | MPNN (graph) | 1.094 | 1.120 | 1.031 |
| MoleculeACE | AFP (graph) | 1.186 | 1.192 | 1.015 |

Note: Results where 'MoleculeACE' is named as the source are retrieved from: https://github.com/molML/MoleculeACE/blob/main/MoleculeACE/Data/results/MoleculeACE_results.csv. Data accessed on November 27, 2025.

| Dataset | XGB + ECFP | XGB + SAE | LSTM (SMILES) | GCN (graph) | GAT (graph) |
|---|---|---|---|---|---|
| CHEMBL1862_Ki | 0.793 | 0.788 | **0.761** | 0.942 | 0.987 |
| CHEMBL1871_Ki | **0.655** | 0.726 | 0.662 | 0.769 | 0.798 |
| CHEMBL2034_Ki | 0.744 | **0.713** | 0.755 | 0.810 | 0.809 |
| CHEMBL2047_EC50 | **0.629** | 0.666 | 0.696 | 0.797 | 0.840 |
| CHEMBL204_Ki | **0.747** | 0.821 | 0.822 | 1.056 | 1.138 |
| CHEMBL2147_Ki | **0.614** | 0.711 | 0.647 | 0.840 | 0.966 |
| CHEMBL214_Ki | **0.664** | 0.740 | 0.723 | 1.007 | 1.051 |
| CHEMBL218_EC50 | **0.714** | 0.757 | 0.748 | 0.928 | 0.957 |
| CHEMBL219_Ki | **0.718** | 0.777 | 0.780 | 1.026 | 0.979 |
| CHEMBL228_Ki | **0.665** | 0.733 | 0.779 | 0.958 | 1.026 |
| CHEMBL231_Ki | **0.764** | 0.774 | 0.809 | 0.878 | 0.991 |
| CHEMBL233_Ki | **0.797** | 0.810 | 0.850 | 1.056 | 1.066 |
| CHEMBL234_Ki | **0.641** | 0.705 | 0.738 | 0.934 | 0.950 |
| CHEMBL235_EC50 | **0.655** | 0.670 | 0.727 | 0.901 | 0.869 |
| CHEMBL236_Ki | **0.696** | 0.798 | 0.812 | 0.942 | 1.002 |
| CHEMBL237_EC50 | **0.757** | 0.815 | 0.783 | 1.132 | 1.103 |
| CHEMBL237_Ki | **0.705** | 0.759 | 0.774 | 1.112 | 1.085 |
| CHEMBL238_Ki | **0.616** | 0.715 | 0.654 | 0.937 | 0.928 |
| CHEMBL239_EC50 | **0.683** | 0.734 | 0.765 | 0.906 | 0.902 |
| CHEMBL244_Ki | **0.722** | 0.828 | 0.800 | 1.075 | 1.088 |
| CHEMBL262_Ki | **0.728** | 0.751 | 0.767 | 0.934 | 0.994 |
| CHEMBL264_Ki | **0.636** | 0.658 | 0.665 | 0.855 | 0.896 |
| CHEMBL2835_Ki | 0.432 | **0.414** | 0.431 | 0.505 | 0.555 |
| CHEMBL287_Ki | 0.741 | **0.711** | 0.791 | 0.886 | 0.947 |
| CHEMBL2971_Ki | **0.612** | 0.671 | 0.689 | 0.781 | 0.803 |
| CHEMBL3979_EC50 | **0.634** | 0.663 | 0.740 | 0.812 | 0.923 |
| CHEMBL4005_Ki | **0.669** | 0.688 | 0.764 | 0.875 | 0.861 |
| CHEMBL4203_Ki | 0.916 | **0.872** | 0.907 | 0.975 | 1.004 |
| CHEMBL4616_EC50 | **0.672** | 0.682 | 0.739 | 0.867 | 0.873 |
| CHEMBL4792_Ki | **0.660** | 0.746 | 0.691 | 0.923 | 1.004 |

Table 13: **Individual MoleculeACE RMSE Scores.** Classical chemistry-inspired features typically outperform both SAE-based predictors and deep learning models, especially on smaller datasets. Our SAE-based features (XGB + SAE) are competitive with, or outperform, deep learning baselines across most targets. Lower is better.

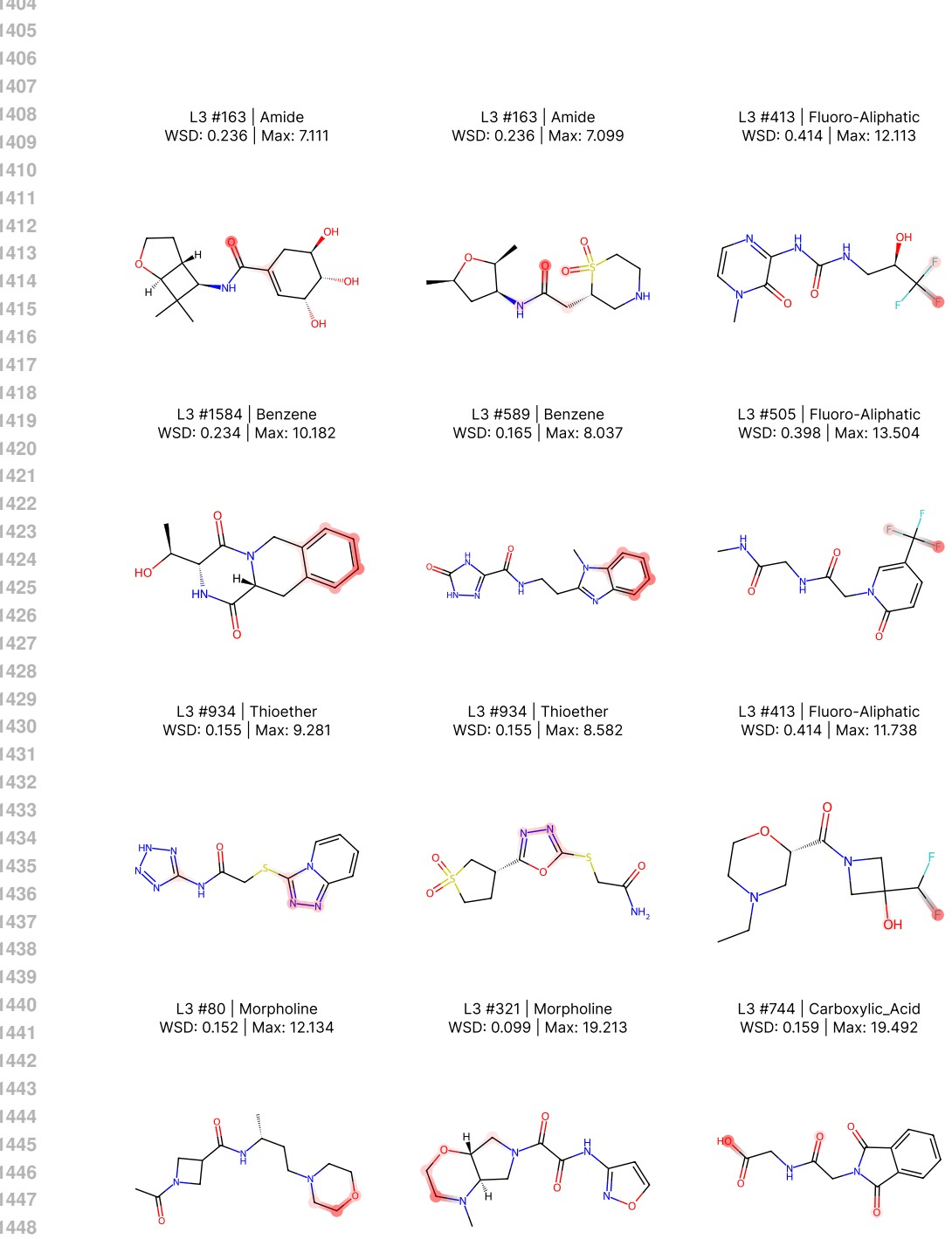

Figure 9: **Example SAE Fragment Activations.** We provide additional SAE activation examples on Layer 3 SAE features from a WSD-based fragment screening on approximately 50K molecules. Red highlights indicate strength of SAE activation on corresponding tokens in molecule.

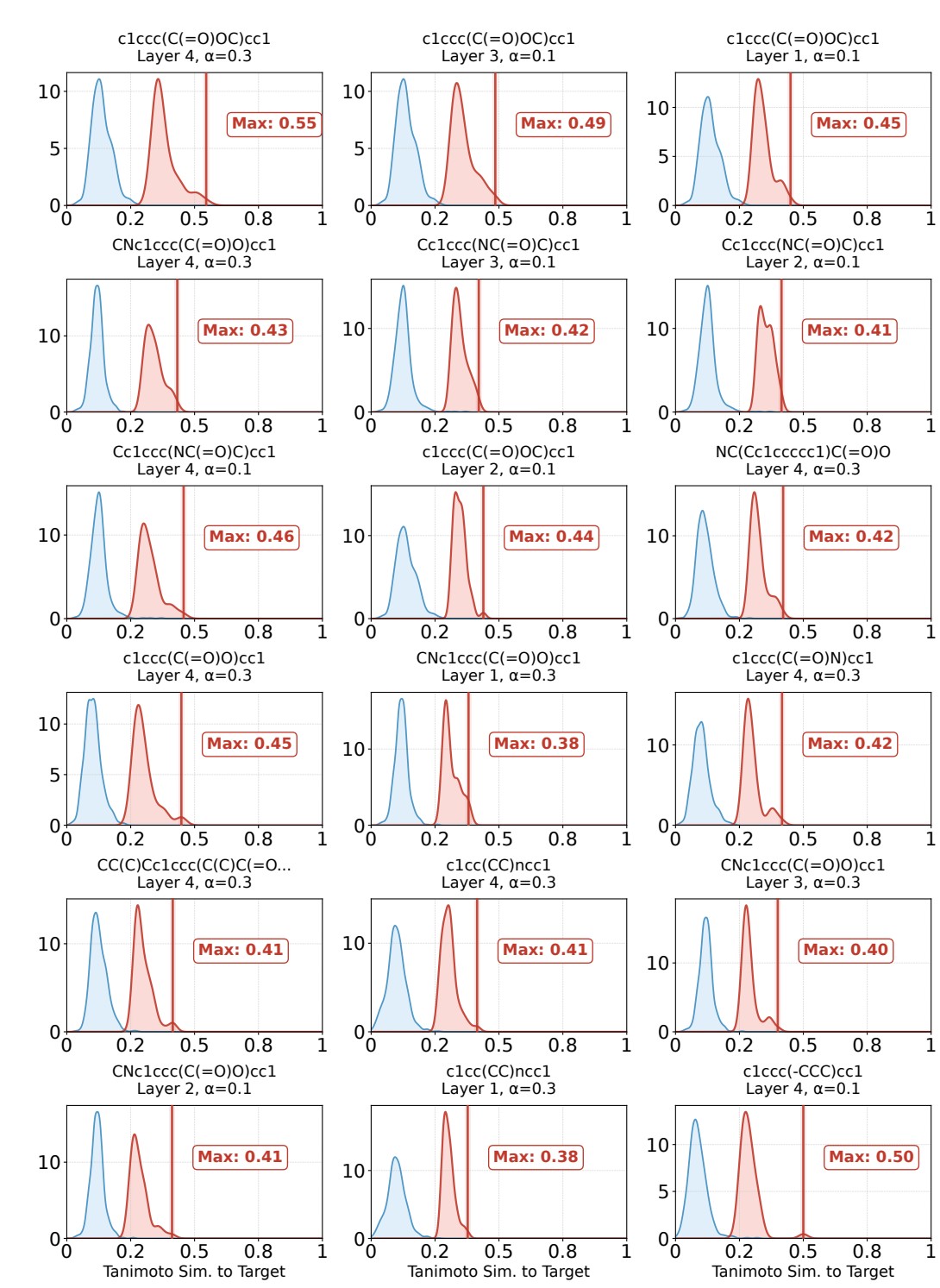

Figure 10: **SAE Features Can Steer Outputs.** KDE comparison of Tanimoto similarity distributions for baseline and steered generations. Each panel corresponds to a different target molecule. Sample molecules from non-conditional generation (**blue**) and SAE steering (**red**) are compared using Tanimoto similarity to the target. Vertical markers denote the maximum similarity achieved under steering for each target. Examples are selected based on highest delta compared to vanilla samples after filtering for experiments where at least half the steered SMILES are valid.

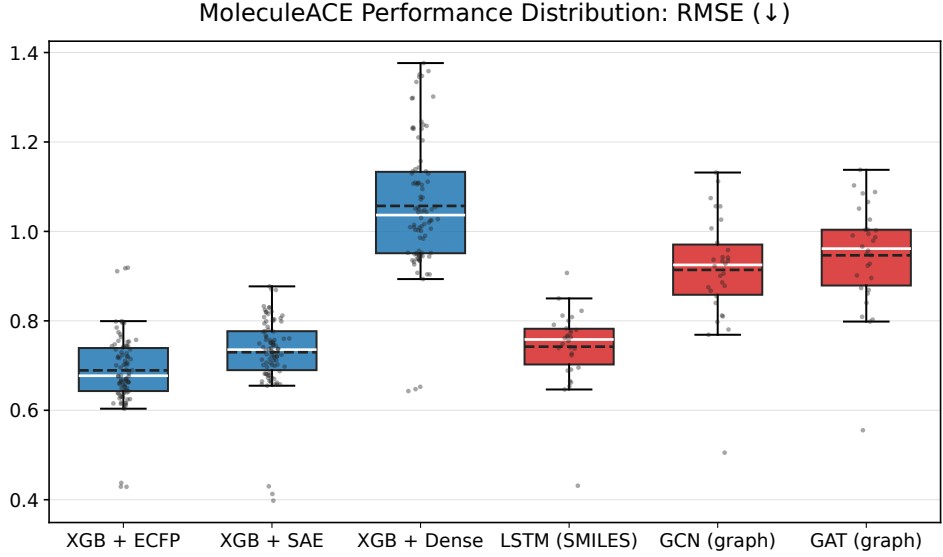

Figure 11: **MoleculeACE Performance Across Methods.** Figure compares RMSE scores achieved by SAE features, traditional fingerprints, and deep learning methods on MoleculeACE endpoints. Blue indicates using the same XGBoost prediction head, with results calculated from three scaffold-split seeds. Red indicates results of popular deep learning models, obtained from from the MoleculeACE repository.

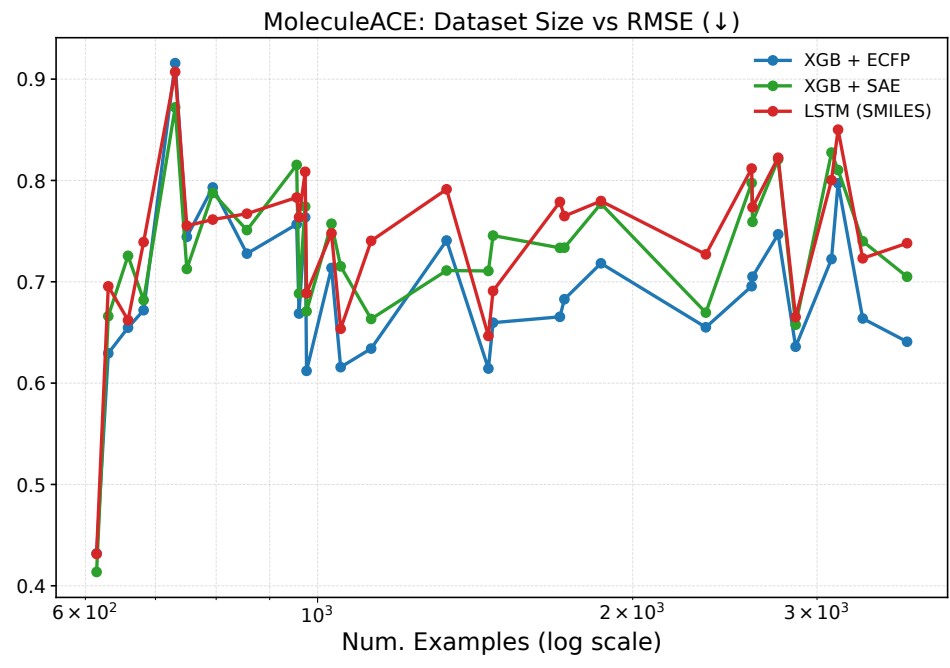

Figure 12: **MoleculeACE Performance Across Dataset Size.** Figure compares RMSE scores achieved by SAE features, ECFP fingerprints, and an LSTM baseline on MoleculeACE endpoints. Results are means across scaffold-split seeds. SAE and ECFP models use the same prediction head.

