# OpenReview forum: "Circuits, Features, and Heuristics in Molecular Transformers"
_ICLR.cc/2026/Conference — Submitted to ICLR 2026_

### Official Review · Reviewer_Qpvr · 2025-10-31

**Soundness:** 2
**Presentation:** 3
**Contribution:** 2
**Rating:** 4
**Confidence:** 4

**Summary:**

The work investigates the internal representations learned by Transformer models when applied to molecular data (SMILES representation). By analyzing attention patterns and feature extraction, the authors aim to uncover how these models capture chemical properties and relationships. The study utilizes sparse autoencoders to offer fresh insights into the interpretability of Transformer architectures in the context of cheminformatics.

I am in the borderline region for this paper. The interpretability perspective is fresh for small molecules, yet, the discovered insights provide limited actionable information for the design of better architectures or for improving downstream performance.

**Strengths:**

1. A fresh perspective on chemical language models is presented using sparse auto-encoders. Interpretable is always interesting in molecular sciences.
2. A broad and in-depth analysis is conducted, and syntactic and semantic patterns are identified.

**Weaknesses:**

1. The analysis is interesting, but it is unclear how these insights can be used to design better architectures or improve downstream performance. The practical implications of the findings are limited.
2. Similar analyses on syntactic patterns were conducted previously as well, in the first molecular transformer paper (MolGPT; pubs.acs.org/doi/10.1021/acs.jcim.1c00600).
3. The patterns are extracted only using a single transformer, for unconditional molecule design and property prediction. It would be interesting to study the same patterns across different models and semantic patterns while generating task-specific molecules, e.g., for bioactive molecule designs.

**Questions:**

1. Can you use the discovered insights to edit transformers? For instance, can the learned circuits be modified to improve performance?
2. Can the authors elaborate on the implementation details of the trained transformer? What is the training data and model configuration? Is the model trained with or without SMILES augmentation?
3. How would SMILES augmentation affect the learned patterns?

---

> ### Author Response · Authors · 2025-11-24
> **Response to Reviewer Qpvr (1/3)**
>
> We sincerely thank you for your valuable feedback and suggestions. To address your concerns regarding downstream performance, actionability, and augmentation, we have conducted new experiments and revised the manuscript. We will briefly respond to your observations and summarize improvements that are most relevant.
>
> ### W1: Actionability and Performance
> **Objection.** *"It is unclear how these insights can be used to design better architectures or improve downstream performance... practical implications are limited."*
>
> **Response.** You are right to point out that some insights may not immediately translate to practical improvements. We feel that a mechanistic characterization of these systems needs to start by confirming these models can develop low-level primitives for validity for further research to be justifiable, and the scope of this work reflects this belief.
>
> However, we believe that investigating whether these systems do more than arbitrary pattern matching can prove useful for future research on its own. Such insights can help decide whether incorporating molecular embeddings from transformers is appropriate in larger drug design pipelienes, and identifying failure modes mechanistically can provide directions for architectural improvements and data augmentation. Such directions could include adopting domain-specific sampling techniques, auxiliary losses, hyperparameter tuning strategies, adding inductive biases to pre-training data, or modifying models to better accomodate circuits for critical mechanisms.
>
> **Property Prediction.** Our original reporting did not include deep learning baselines as the primary goal was to assess whether sparsity can improve representations and whether unsupervised features can match a well-established rigid baseline. We extended our reporting on classification task performance to include popular deep learning models from the TDC leaderboard, displayed in *Table 5* in the Appendix.
>
> * **Against Supervised Baselines:** On several tasks, SAE and SAE+ECFP features match or beat the performance of Chemprop-RDKit, an established GNN baseline that also incorporates chemical features. On hERG cardiotoxicity, SAE features outperform the standard supervised baseline Chemprop (+0.009) and significantly outperform ECFP (+0.057). As detailed in **Section 5.1**, adding SAE features can bridge the gap between fingerprints and performant GNNs.
>
>
> **Table R1: Unsupervised Features vs. Supervised Baselines (TDC Leaderboard)**
>
> *Comparison of our SAE features against a popular GNN-based model on selected ADMET tasks.*
>
> | Method | Supervision | ROC-AUC (hERG) | ROC-AUC (Pgp) |
> | :--- | :--- | :--- | :--- |
> | **ECFP** (Fingerprint) | None | 0.792 | 0.895 |
> | **Chemprop** (GNN) | Supervised | 0.840 | 0.886 |
> | **SAE Features (Ours)** | **Unsupervised** | **0.849** | **0.922** |
> |**MapLight + GNN** |	Supervised |	*0.880* |		*0.938* |
>
> We also note that these results do not mean that SAE activations should be preferred over supervised, more structured SOTA approaches, but we do believe they support such these features can improve predictive performance, especially for classification.
>
> The newly added experiments on SAE steering, as described later in our response, also indicate that SAE features can impose usable constraints on search space for unconditional autoregressive generation.
>
> Message continues.

---

> ### Author Response · Authors · 2025-11-24
> **Response to Reviewer Qpvr (2/3)**
>
> ### W2: Novelty vs. Related Work
> **Objection.** *"Similar analyses on syntactic patterns were conducted previously... in the first molecular transformer paper (MolGPT)."*
>
> **Response.** The authors of MolGPT analyze saliency maps to assess interpretability and identify patterns related to valence, branch balancing, and ring matching. There is indeed an overlap in the subjects of investigation, but we argue that our research complements and significantly extends these findings. We provide a mechanistic analysis to identify **circuits** via intervention (ablation and steering) and retrieve sparse, disentangled **features** that activate in specific chemical settings, which presents a more granular picture.
> - **Local Intervention vs. Attention:** As shown in **Section 3.1** (Ring/Branch Circuits), we distinguish between heads that "attend" (pointers) and heads that "write" (execution). We show that while many heads look at ring numbers (as seen in MolGPT), only specific heads are load-bearing. The newly added experiment shows that ablating the "writer" also causes validity to drop to 25.4%, whereas ablating "pointers" has modest effect.
> - **Semantic Features:** To our knowledge, this work is the first to utilize sparse dictionary learning for small molecules to obtain chemical features from the residual stream. We describe a screening process for finding fragment-specific neurons to accelerate feature annotation and discovery, which we believe improves scalability compared to manually analyzing saliency maps or activation patterns.
> - **Utility:** We believe sparse feature dictionaries are more easily utilized in practice than attention maps, and the extracted features can be used to augment larger pipelines. SAE features also expose a lever to guide non-conditional generative transformers in exploring chemical space.
>
> We appreciate your observation regarding MolGPT and we have extended the Related Work section in **Appendix A** to better contextualize our contributions and better address findings from adjacent research.
>
>
> ### W3: Scope of Investigation
>
> **Objection.** *"The patterns are extracted only using a single transformer, for unconditional molecule design and property prediction. It would be interesting to study the same patterns across different models and semantic patterns..."*
>
> **Response.** We fully agree with your comment on the importance of continuing this line of work to cover different architectures and training objectives. While the scope of our study could not extend to conditional generators and different transformer variants, such experiments would significantly impact our understanding of molecular transformers. Comparing mechanisms emerging when a model is required to match explicit conditions versus solely optimizing chemical space traversal would provide much needed clarity. Circuits or SAE neurons that regulate these functions could also expose an additional control surface for such models. We have added comments on possible future work in the newly created Appendix A.2.
>
> **Circuit Robustness.** To ease uncertainty, we added bootstrapped CI's for valence probes and ran additional ablations for syntax circuits in **Section 3**. The results solidify that ring and branch heads are involved in information transport and maintaining validity, and also verify that the model has a persistent mid-network direction that encodes valence, and modulating it provides a subtle but statistically significant shift in bond token predictions, increasing logits in the *"high valence"* direction.
>
> **SAE Universality.** We added a new experiment to assess feature overlap between molecular SAEs with different dictionary sizes and see how many features could be artifacts of a specific training run. Our analysis, described in **Appendix E.2**, measures feature stability across SAEs with 2048, 4096, and 8192 latents at every layer of our transformer. We calculate maximal cosine alignment on SAE decoder weights and also report a random baseline for control.
>
> While Layer 0 activity varies significantly, Layer 1 demonstrates an exceptionally high degree of cross-SAE alignment, with a mean MCS of 0.940 between the 8x and 16x autoencoders and over 80% of the features in the smaller dictionary are recovered with strict fidelity in the larger dictionary. In later layers, feature similarity remains high, but strictly overlapping features are rare, implying that these SAEs learn similar representations at different sizes, but neurons in larger SAEs are specialized relatives of counterparts in smaller dictionaries, similar to how feature splitting would behave in a natural language setting.
>
> Message continues.

---

> > ### Author Response · Authors · 2025-11-24
> > **Response to Reviewer Qpvr (3/3)**
> >
> > ### Q1: Circuit and Feature Interventions
> >
> > **Question.** *"Can you use the discovered insights to edit transformers? For instance, can the learned circuits be modified to improve performance?"*
> >
> > **Answer.** We share your interest in activation editing to modulate non-conditional transformer behavior. As for valence and syntax circuits, we believe interventions on these specialized components could have use-cases in edge scenarios where long-range dependencies are important, or when data/parameter scaling is infeasible. As the autoregressive training objective forces transformers to explore a large chemical space with a fixed set of parameters, out-of-distribution prompts or high sampling temperatures can quickly tank validity.A principled way to modulate syntax circuits may help maintain output quality.
> >
> > As for editing sparse activations, we now share preliminary results that indicate SAE dictionaries can guide transformer trajectories during inference.
> >
> > **Feature Steering.** To test if SAE features are causal "control knobs," we implemented a steering experiment (described in **Section 5.2** and **Appendix F.4**). We extracted the latent representation of a target drug-like molecule and injected a scaled version of this vector back into the residual stream during autoregressive generation.
> > *   **Result:** This intervention effectively steered the model toward the target region of chemical space. We observed a consistent increase in Tanimoto similarity between the generated samples and the target molecule compared to the unsteered baseline.
> > *   **Implication:** This confirms that the features identified are not merely correlational but can be used to bias the generator toward specific chemical substructures without retraining the model. This addresses your question regarding "editing" the model to improve utility.
> >
> > We note that the results indicate an ability to direct generation towards desired regions in the search space, not as a way to obtain a sophisticated conditional model for free. However, this indicates that refined steering approaches can be utilized effectively in such scenarios.
> >
> > ### Q2, Q3: SMILES Augmentation
> >
> > **Question.** *"Is the model trained with or without SMILES augmentation? How would SMILES augmentation affect the learned patterns?*
> >
> > **Answer.**
> > We added a detailed description of the implementation details including the models, datasets, and experimental setup. As detailed in **Section 2.2** and **Appendix F.3**, the transformer was trained on *canonical* SMILES strings derived from the ZINC20 lead-like subset (approx. 244M training samples). We chose canonical SMILES to ensure that specific tokens have a consistent positional context, which is crucial for establishing a clean baseline for mechanistic analysis.
> >
> > **Effect of Augmentation (Robustness).** To answer your question about how augmentation affects learned patterns, we conducted a new experiment (**Section 4.1**) where we fed non-canonical (augmented) SMILES of the same molecules into the *canonical-trained* model and measured feature stability.
> > *   **Early Layers (Local Invariance):** In Layers 0 and 1, SAE features exhibit high robustness (Jaccard similarity $\approx 0.97$, Cosine similarity $\approx 0.94$). This indicates that the model's recognition of atoms, bonds, and immediate neighbors is **invariant** to the specific SMILES permutation; it recognizes the chemistry regardless of the string order.
> > *   **Deep Layers (Path Dependence):** In deeper layers (L3-L5), stability drops significantly. This indicates that while the model may use early layers for syntactic parsing and chemical validity, the *generation algorithm* (deep layers) is highly path-dependent, and a large share of these SAE features seem more involved in generation rather than providing abstractions.
> >
> > We believe this distinction—that early features are chemically robust while deeper features are syntax/path-dependent—is a valuable insight that would have been obscured without the new data, and we thank you for prompting this line of investigation.
> >
> >
> > ### Conclusion
> >
> > We hope these additional experiments on downstream performance, feature steering, and robustness address your concerns regarding the actionability and scope of our work. We believe the revised manuscript offers a stronger case for the utility of mechanistic interpretability in molecular machine learning. We thank you for reviewing our work and we are happy to answer any further questions you might have.
> >
> >
> > Best regards, The Authors

---

> ### Comment · Reviewer_Qpvr · 2025-11-26
> **Response I to Authors**
>
> I thank the authors for answering my comments and questions with experiments. I find their work truly fresh and interesting. Most of my concerns have been addressed.
>
> I will be happy to increase my score to 6, **if** the authors can include the connection with MolGPT in the *introduction* to make it clear that similar analysis has been done before. I believe MolGPT work deserves more credit than being in the appendix.
>
> The reason why I am not increasing my score to 8 is the use of the TDC benchmark. TDC is a highly-criticized [1], low-quality dataset that should be abandoned by the community. If the authors can include results on a higher-quality benchmark (such as [2,3]; could simply be a 3-task subset) in the main text, and move TDC to the appendix, I will be happy to further increase my score to 8.
>
> [1] http://practicalcheminformatics.blogspot.com/2023/08/we-need-better-benchmarks-for-machine.html
>
> [2] https://pubs.acs.org/doi/10.1021/acs.jcim.3c00160
>
> [3] https://pubs.acs.org/doi/10.1021/acs.jcim.2c01073

---

> > ### Author Response · Authors · 2025-11-28
> > **Response to Reviewer Qpvr (4/4)**
> >
> > Thank you for your feedback. We are glad you found our additions valuable, and the additional suggestions are much appreciated. We revised the manuscript and added new experiments to accommodate your requests. Here, we take the opportunity to provide a brief summary.
> >
> > **Highlighting Previous Work.** We value your feedback concerning the impact of seminal work on molecular transformer interpretability. To better contextualize our study and clarify its value proposition, the Introduction has been revised to explicitly highlight the interpretability findings of MolGPT.
> >
> >
> > **Extended Downstream Results.** We also appreciate your input on dataset reliability. We added new experiments on public, experimentally backed endpoints and revised **Section 5.1** to focus on these results and deferred the treatment of TDC performance to the appendix.
> >
> > **Evaluation on cADME [1]:** We benchmarked on four public cADME endpoints. As shown below, unsupervised SAE features consistently outperform the fingerprint baselines (ECFP/FCFP), though they trail the fully supervised Graph Neural Networks (MPNN).
> >
> > | Task | ECFP | FCFP4 + rdMD | MPNN1 | MPNN2 | SAE (Ours) |
> > | :--- | :--- | :--- | :--- | :--- | :--- |
> > | HLM | 0.578 | 0.575 | 0.676 | 0.714 | **0.617** |
> > | MDR1 | 0.656 | 0.643 | 0.666 | 0.735 | **0.676** |
> > | RLM | 0.605 | 0.575 | 0.727 | 0.735 | **0.659** |
> > | Solubility | 0.461 | 0.502 | 0.618 | 0.638 | **0.513** |
> >
> >
> > **Evaluation on MoleculeACE [2]:**
> > We evaluated the full set of 30 activity-cliff endpoints and compared unsupervised SAE features against the supervised deep learning baselines reported in the original study.
> >
> > | Source | Method | RMSE | RMSE\_cliff |
> > | :--- | :--- | :--- | :--- |
> > | Ours | XGB + ECFP Fingerprints | 0.689 | 0.771 |
> > | **Ours** | **XGB + SAE Features** | **0.730** | **0.810** |
> > | [2] | LSTM (SMILES) | 0.742 | 0.871 |
> > | [2] | GCN (Graph) | 0.914 | 0.974 |
> >
> >
> > **Interpretation:**
> > 1.  **cADME:** SAE features close a significant portion of the gap between fingerprints and supervised GNNs using only self-supervised pretraining.
> > 2.  **MoleculeACE:** Consistent with the findings in [2], expert-engineered fingerprints (ECFP) remain a very strong baseline on activity cliffs. However, our **SAE features outperform the supervised deep learning baselines (LSTM, GCN)** reported in the original paper.
> >
> > We believe these results on experimentally backed endpoints confirm that SAE features capture chemically relevant information that can translate to downstream performance, both in cases where large deep learning architectures dominate and in cases where classical featurization prevails.
> >
> > We are incredibly grateful for the specific benchmark suggestions, which have significantly strengthened the empirical validation of our paper.
> >
> > Best regards, The Authors
> >
> >
> > ***
> >
> > [1] Fang, C., et al. (2023). Prospective Validation of Machine Learning Algorithms... *JCIM*.
> >
> > [2] van Tilborg, D., et al. (2022). Exposing the Limitations of Molecular Machine Learning with Activity Cliffs. *JCIM*.

---

### Official Review · Reviewer_wRsj · 2025-11-01

**Soundness:** 3
**Presentation:** 3
**Contribution:** 2
**Rating:** 4
**Confidence:** 5

**Summary:**

The focus of the paper is a mechanistic analysis of autoregressive transformer trained on drug-like molecules that elucidates computational structure behind chemical inference.

**Strengths:**

The paper addressed an important task of understanding how deep learning architectures mechanistically solve inference tasks in a way that is congruent with the scientific structure of the domain and can be assessed by subject matter experts.

Application of sparse autoencoder clearly helps with feature engineering for pharmacological tasks.

**Weaknesses:**

The authors are unreasonably generous with the term "chemical reasoning". Neither the model that they chose nor analysis that they performed elevate to reasoning level - only good old correlations.

it's not obvious what exactly one gains from the mechanistic analysis performed on SMILES. SMILES have peculiar syntax - any model that handle SMILES has to be able to deal with it. We know that transformers can deal with SMILES - what has changed in our understanding once we learned which head tracks the positions of opening and closing parentheses? it would be interesting to see analysis of this type on some non-trivial abstractions, but parsing syntax of the primary representation, such as SMILES, is not enlightening.

If "chemical reasoning" comes down to capturing SMILES syntax, it is not a meaningful bar to cross for ICLR paper in 2025.

**Questions:**

Please unpack the phrase: "Understanding how these models encode chemical knowledge enables more targeted interventions, better failure diagnosis, and principled approaches to model improvement". None of the "enabled" items here immediately follows from the study.

---

> ### Author Response · Authors · 2025-11-25
> **Response to Reviewer wRsj (1/2)**
>
> We sincerely thank you for your review and useful feedback. We will briefly address your comments and summarize additions to our study that you may find valuable.
>
>
> ### W1: On Terminology
>
> **Objection.** *"The authors are unreasonably generous with the term "chemical reasoning". Neither the model that they chose nor analysis that they performed elevate to reasoning level - only good old correlations."*
>
> **Response.** We appreciate the critique regarding terminology. We were indeed liberal with the term 'Chemical Reasoning', which was used as a shorthand for how a sequential model assembles valid representations of molecules to satisfy certain chemical criteria. To avoid anthropomorphizing these models, we replaced all instances with substitutes that do not imply a level of cognitive agency.
>
> Your concerns regarding these abilities are absolutely warranted, which is why we believe investigating mechanisms that maintain validity is a necessary  prerequisite to higher-level analysis.
>
> ### W2: Utility of Syntax Circuits
>
>
> **Objection.** *"SMILES have peculiar syntax - any model that handle SMILES has to be able to deal with it... what has changed in our understanding...?"*
>
> **Response.** Our core objective is to characterize *how* SMILES are generated in autoregressive transformers. While SMILES parsing is syntactically solved, how transformers handle this without a stack memory is unclear, and even sophisticated models can make mistakes. Similar investigations are an active area of interpretability research for multiple formal grammars (e.g., Dyck-k languages), and this issue serves as a tractable first step to confirm fundamental circuits are present before moving to complex chemical properties.
>
> We believe that studying syntax also provides practical insights. Misplaced or forgotten ring digits and closing parentheses still trick transformers, decreasing sample validity. Robust circuits for these functions can be especially useful for long or low-likelihood sequences, and a baseline characterization can point to architectural changes or data augmentation with adversarial examples that may help accomodate these.
>
> **Our Contribution.** Current understanding of autoregressive SMILES transformers remains surface-level. Most works treat these models as opaque systems and analysis is often limited to dissecting ad-hoc attention patterns, which introduces uncertainty.
>
> This study presents a **mechanistic** treatment of molecular transformers. We point to lower-level computational units  that are involved in syntactic and chemical validity and observe their behavior under intervention.
>
> ### W3: Higher-level Circuits and Features
>
> **Objection.** *"...it would be interesting to see analysis of this type on some non-trivial abstractions... If "chemical reasoning" comes down to capturing SMILES syntax, it is not a meaningful bar to cross..."*
>
> **Response.** We agree that abstract chemical concepts may be more interesting to study and are likely to be more useful for practicioners. Apart from syntactic circuits, the  original manuscript included an analysis of valence representation and described screening methodology for finding chemical patterns in SAE features. We have added new details and experiments to expand on these results.
>
> The revised manuscript now expands the analysis of higher-level abstractions:
>
> - **Chemically meaningful SAE features.** Section 4 has been expanded with a more systematic evaluation of SAE-derived features for SMARTS queries. Figure 4 and Table 8 show that sparse features exhibit substantially higher specificity than dense residual or PCA bases and yield near-monosemantic units in some cases, and results confirm that many features align with structure-level motifs and are not tied to SMILES quirks.
>
> - **Downstream validation.** We previously showed that unsupervised SAE features consistently outperform dense transformer embeddings and PCA on multiple property prediction. We now show that for many TDC classification endpoints (e.g., hERG, P-gp), SAE features also match or exceed supervised GNN baselines, and adding SAE input features can bridge the gap between fingerprints and more powerful supervised architectures. This suggests that sparse features can capture or approximate pharmacologically relevant signals beyond syntactic correctness.
>
> - **Preliminary control via feature steering.** Section 5.2 adds an illustration of how sparse features can be used to steer generation. Using SAE activations, we can inject biases at inference to steer generation toward a more constrained regions of chemical space. In most cases, we observe a significant increase in sample similarity between generated sequences, and for a number of drug-related target molecules, this also notably increases sample similarity with the target molecule.
>
> Message continues.

---

> > ### Author Response · Authors · 2025-11-25
> > **Response to Reviewer wRsj (2/2)**
> >
> > ### Q1: On “targeted interventions, failure diagnosis, and principled model improvement”
> >
> > **Question.** *“Please unpack the phrase: ‘Understanding how these models encode chemical knowledge enables more targeted interventions, better failure diagnosis, and principled approaches to model improvement’. None of the ‘enabled’ items here immediately follows from the study.”*
> >
> > **Response.** We agree that, as originally written, this sentence compressed several distinct claims that may not directly follow from our results. In the revision, we have (i) refined phrasing in the Conclusion to avoid implying fully realized applications, and (ii) made explicit which parts are demonstrated versus prospective.
> >
> > 1. **What we demonstrate in the paper.**
> >
> >    - **Targeted interventions.** We perform local causal interventions on specific components (ablating syntax heads, steering along the valence direction, and injecting SAE latents) and quantify their effect on validity and bond-order logits. These are direct, model-internal interventions motivated by the identified circuits and directions.
> >    - **Failure characterization.** Head-level ablations and event-specific margins already let us characterize *how* validity fails (e.g., ring closures collapse when the writer head is removed, whereas branch balancing is more robust). Similarly, valence steering reveals that the model’s willingness to emit higher-order bonds can be modulated.
> >    - **Model improvement signals.** The downstream TDC experiments show that molecular SAE features improve or complement standard baselines. This suggests that understanding and re-expressing internal representations can yield practically useful feature spaces, but we stop short of proposing concrete architecture changes in the current work.
> >
> > 2. **How these results motivate future work.**
> >
> >    Rather than making a claim to immediate applications, we now connect the above findings to promising directions for further research. We believe that expanding on our work and refining the described approaches can provide further practical value to interpreting molecular transformers. These suggestions are found in more detail in the newly created Appendix A.2.
> >
> > ### Conclusion
> >
> > Thank you for the time and effort spent reviewing our work. Your suggestions helped us clarify key terminology and your comments on the relevance of this line of research help us revise the document to better contextualize our work.
> >
> > We have added further details to experiments and also conducted new analyses for SAE robustness, universality, and also added preliminary results on using these activations to steer models during inference. We believe these additions also make a stronger case for adopting mechanistic approaches to deep learning in computational chemistry.
> >
> > We hope to have addressed your concerns with this response and the revisions to the manuscript. We are happy to answer further questions or respond to any remaining objections.
> >
> > Best regards, The Authors

---

### Official Review · Reviewer_UkWj · 2025-11-01

**Soundness:** 2
**Presentation:** 2
**Contribution:** 2
**Rating:** 4
**Confidence:** 2

**Summary:**

This paper conducts a mechanistic interpretability study of autoregressive molecular transformers. It identifies attention head circuits responsible for SMILES syntactic correctness, discovers a linear residual feature encoding valence capacity that influences bond order prediction, and uses sparse autoencoders to extract chemically meaningful features. The authors further validate these insights through downstream molecular property prediction tasks, showing that SAE-derived features provide competitive or complementary predictive performance.

**Strengths:**

1. The work focuses on internal mechanisms related to SMILES syntax handling (e.g., ring closure and branch matching), which is directly relevant to the problem of generating syntactically valid molecules. This aligns with concerns in molecular generation research, where validity constraints are important.

2. The identification of a linearly readable representation related to valence capacity in the residual stream is interesting and may offer conceptual hints for improving structural consistency in generative models.

3. The use of sparse autoencoders to decompose intermediate representations into more interpretable latent features is an exploratory direction that could be valuable for connecting molecular language models to chemically meaningful concepts.

**Weaknesses:**

1. Limited connection to practical improvements in molecular generation

Although the paper reveals mechanisms for syntax and valence representation, it is not yet clear how these findings could be used to improve molecular generative performance (e.g., validity, novelty, or property-aware design). The interpretability results currently feel more diagnostic than actionable.

2. Interpretability of SAE-derived features varies considerably

While some features appear to align with recognizable chemical motifs (e.g., urea groups), a substantial portion remains difficult to interpret or validate. The paper also acknowledges this, which makes it unclear how reliably these features can guide model understanding or downstream applications.

3. Downstream evaluation suggests modest and inconsistent performance improvements

On some TDC tasks, SAE features are competitive or complementary with ECFP, but overall improvements are relatively small and task-dependent. It is difficult to determine whether the observed performance differences justify the additional complexity introduced by SAE analysis.

4. Comparisons do not include more expressive molecular representation methods

The experimental evaluation compares primarily against ECFP and the raw LM embeddings. However, recent advances in graph neural networks and 3D-aware models have shown strong performance in molecular property prediction. Without comparison to these stronger baselines, it is hard to contextualize the practical value of the proposed interpretability findings.

5. Causality interpretation remains preliminary

The valence budgeting direction in the residual space is evaluated via linear probing and steering, which provides suggestive but not conclusive causal evidence. The paper itself is careful about this, but it still means the main chemical reasoning claims remain partly correlational.

**Questions:**

- Do SAE features provide measurable benefit beyond ECFP or learned embeddings when used in larger-scale or more challenging prediction settings?
It would help to better understand whether the value of SAE features extends beyond interpretability.

- Can the authors provide any quantitative measure or estimate of how many SAE features are consistently interpretable across runs?
This would help clarify how robust and generalizable the feature discovery pipeline is.

---

> ### Author Response · Authors · 2025-11-23
> **Response to Reviewer UkWj (1/2)**
>
> We thank you for the detailed assessment, and for identifying key limitations of the original manuscript regarding actionability, baselines, and interpretability. We have added new experiments to address your concerns and expanded on key details to better contextualize our findings. We take this opportunity to briefly respond to your questions and summarize the main revisions to the manuscript.
>
> ### W1: Unclear Real-world Utility
>
> **Objection.** "It is not yet clear how these findings could be used to improve molecular generative performance... interpretability results currently feel more diagnostic than actionable."
>
> **Response.** We believe mechanistic interpretability serves three distinct roles in computational chemistry: (1) *Transparency* (describing internal mechanisms and biases), (2) *Guidance* (providing new directions for architectural changes, data augmentation, etc.), and (3) *Enhancement* (enabling better feature extraction and sampling).
>
> We acknowledge that the treatment of syntax and valence circuits in *Section 3* is directed at transparency and diagnostics. However, we believe determining whether mechanisms for core elements of correctness arise during training, or if the model merely memorizes patterns, is a necessary base layer for trusting these models in generative loops.
>
> Our results from downstream property prediction experiments indicate that mechanistic insights can be used to improve the quality of representations that come from these autoregressive models, and can also guide clarity on how these systems generate molecules. For example a newly implemented experiment produced  results that we found counterintuitive at first.
>
> Architectural Insights from SMILES Augmentation. Based on Reviewer Qpvr's comments, we analyzed SAE features under SMILES randomization. The results in **Section 4.1** suggest that the causal language modeling objective, especially with canonical molecules, can make late layers develop more context-specific features. We believe determining how data augmentation (both for transformers and SAEs) could affect this is an interesting direction for future work. Findings can also help practicioners decide whether autoregressive features are robust enough for their use-case. If larger networks develop more stable mid-layer features, it would indicate that scaling model size for better representations can be logical even after standard metrics saturate.
>
>
> ### W2: Choice of Baselines
>
> **Objection.** "Comparisons do not include more expressive molecular representation methods (e.g., GNNs)... improvements are relatively small and task-dependent."
>
> **Response.** Our main objective in *Section 5.1* is to answer whether feature sparsity provides valuable information compared to vanilla residual-stream neurons. However, we agree that context is needed regarding absolute performance. We have addressed this by comparing our results against the *Therapeutics Data Commons (TDC) Leaderboard*.
>
> While we do not claim SOTA against specialized supervised models, our *unsupervised* SAE features perform competitively with, and effectively bridge the gap to, supervised baselines.
>
> **Table R2: Bridging the Gap on Safety Pharmacology (hERG)**
>
> *Comparison of our Unsupervised Features against Baselines and External Supervised Models (TDC Leaderboard).*
>
> | Method | Supervision | ROC-AUC (hERG) | $\Delta$ vs ECFP (hERG) | ROC-AUC (Pgp) | $\Delta$ vs ECFP (Pgp) |
> | :--- | :--- | :--- | :--- | :--- |:--- |
> | **ECFP** (Baseline) | None (Fixed) | 0.792 | - | 0.895 | - |
> | **Chemprop-RDKit** | *Supervised* | 0.840 | +0.048 | 0.886 | -0.009 |
> | **SAE** (Ours) | *Unsupervised* | **0.849** | **+0.057** | **0.922** | **+0.027** |
> | **MapLight + GNN** |  *Supervised* | 0.880 | +0.088 | 0.938 | +0.043 |
>
> * **Beating Supervised Baselines:** On hERG, our SAE features ($0.849$) outperform the standard supervised baseline Chemprop ($0.840$). On Pgp Inhibition, Chemprop actually underperforms simple fingerprints ($\Delta -0.009$), whereas our unsupervised SAE features successfully extract signal to improve over the baseline ($\Delta +0.027$).
> * **Gap Recovery:** While specialized architectures like MapLight remain the state-of-the-art, our method bridges a consistent majority of the performance gap between fingerprints and MapLight-GNN ($\sim65$% recovery on hERG and $\sim63$% on Pgp).
> * **Implication:** This validates that the Transformer has learned complex biological signals that are superior to standard fingerprints and competitive with supervised graph representations, even without access to toxicity labels during training.
>
> Message continues.

---

> > ### Author Response · Authors · 2025-11-23
> > **Response to Reviewer UkWj (2/2)**
> >
> > ### W3: SAE Stability
> > **Objection.** "A substantial portion remains difficult to interpret... Can the authors provide any quantitative measure of how many SAE features are consistently interpretable across runs?"
> >
> > **Response.** We acknowledge that interpreting features remains challenging.
> > The main difficulty is not that the majority of SAE neurons are incoherent (as is often the case for natural language), but that many of them produce highly specialized activation patterns that do not immediately translate to common chemical artifacts. To address this, we revised *Section 4* to rely less on manual inspection and performed new experiments to quantify *quality* and *consistency*.
> >
> > **Feature Specificity (Quality).** We implemented score to test wheter SAE features detect chemical fragments. We compared SAE features against Dense (residual-stream transformer), Random, and PCA baselines.
> >
> > **Table R3: Fragment Specificity (Max WSD Scores)**
> >
> > *Data derived from Table 8 in the revised manuscript. Higher is better.*
> >
> > | Fragment Type | PCA Baseline | **SAE (Ours)** |
> > | :--- | :--- | :--- |
> > | **Nitrile** (-C#N) | 0.02 | **0.44** |
> > | **Fluoro-Aliphatic** | 0.09 | **0.42**
> > | **Sulfonamide** | 0.14 | **0.29** |
> > | **Ketone** (C=O) | **0.44** | 0.18 |
> >
> > *Conclusion:* For most functional groups (e.g., Nitriles, Sulfonamides), SAE features provide significantly higher specificity. For simple, high-frequency atomic tokens like Ketones, linear methods (PCA) suffice, whereas SAEs tend to split these into more granular, context-dependent features.
> >
> > **Feature Universality (Consistency).** To answer your question on consistency, we compared features learned by SAEs of different sizes ($4x \rightarrow 16x$) trained on the same data (*Figure 3* and *Section E.2*).
> > * *Finding:* *Layer 1* features are exceptionally stable, with *$>79$% strict recovery* (cosine similarity $> 0.9$) between models.
> > * *Implication:* While deeper layers exhibit "feature splitting," the chemical primitives in early layers are highly reproducible and not random artifacts of training.
> >
> >
> > ### W4: Valence Claims
> >
> > **Objection.** "Valence budgeting... suggestive but not conclusive causal evidence."
> >
> > **Response.** We agree that "reasoning" was an unfortunate term to use and have softened the language in the revised document. We must also clarify that the extracted valence vector does not constitue a singular explanation for all of the model's valence-related behavior. However, we maintain that our evidence exceeds simple correlation, and that our findings indicate that transformers *can* access valence information during inference.
> >
> > Regarding causality, *Figure 2b* demonstrates that intervening on the discovered direction ($x \leftarrow x + \alpha \hat{w}$) directly modulates the model's output distribution for bond tokens.
> > * *Robustness:* We have updated Figure 2b with **$95$% bootstrap confidence intervals** ($N=1000$). The results show a monotonic, statistically significant increase in double/triple bond probability as $\alpha$ increases.
> > * *Mechanism:* While we admit the mechanism is likely *distributed* (we find no evidence that a single "valence neuron" exists), the results show that the *direction* in the residual stream acts as a writable variable for this property.
> >
> > ### Conclusion
> > We hope to have sufficiently answered your questions regarding our work. We believe the additional quantitative analyses—specifically the *Augmentation Robustness* (Table R1), *Feature Stability* (Figure 3), and *TDC Benchmarking* (Table R2)—directly address your concerns regarding actionability and reliability.
> >
> > We thank you for your time spent reviewing this work and are happy to answer further questions.
> >
> > Best regards, Authors

---

### Official Review · Reviewer_SzMd · 2025-11-10

**Soundness:** 2
**Presentation:** 4
**Contribution:** 3
**Rating:** 8
**Confidence:** 4

**Summary:**

This study explores how to interpret what Chemical Language Models (CLMs) trained on SMIELS are actually learning. Then, using Sparse AutoEncoders (SAEs) they extract higher level features that are useful for different predictive task.

**Strengths:**

1. The paper provides a novel perspective on mechanistic interpretability of CLMs focusing on syntactic rules like: ring opening and closing or valence budgeting.
2. Although the use of SAE for interpretability in LMs is not new, the approach here proposed leveraging SAE with SMAER patterns is novel to the best of my knowledge.
3. The analysis of the SAE features is fair and the limitations are properly acknowledged, and it is interesting that there is still need for human expert selection; despite the role of the automatic pipeline in identifying promising candidate features.
4. The experimental validation in the TDC ADMET predictive tasks provide important contextualization for the benefits, or more accurately, the informativeness of the SAE features. The results with SAE alone are convincing when compare to the LM or LM-PCA approaches.

**Weaknesses:**

1. Statistical robustness: the analysis of causal impact in sections 3.1 and 3.2 lack an appropriate statistical analysis. I'm not entirely sure what statistical test would be the most appropriate, but considering the number of implicit hypotheses that are being tested, I think that it is important to ensure that the results are not spurious. Similarly, Table 1 and Figure 2, should contain dispersion metrics (or true confidence intervals) calculated through different samples to show the uncertainty of the results.

**Questions:**

None

---

> ### Author Response · Authors · 2025-11-23
> **Response to Reviewer SzMd**
>
> We sincerely thank you for the positive assessment. We are glad you found our analysis useful and we appreciate your suggestions to improve our work. We extended our experiments to include dispersion metrics and confidence intervals you requested. Here, we summarize the relevant changes in the revised manuscript.
>
> ### W1: Statistical Robustness and Uncertainty
>
> **Objection.** "The analysis of causal impact in sections 3.1 and 3.2 lack an appropriate statistical analysis... Table 1 and Figure 2 should contain dispersion metrics (or true confidence intervals)..."
>
> **Response.** We agree that reporting single-point estimates for causal interventions may have obscured variability. To address this, we have updated our results with bootstrap confidence intervals and ran a new experiment to assess the impact of syntax-related heads on sample validity.
>
> **Probe Accuracy.** We computed 95% confidence intervals for the linear probe accuracy across all layers using stratified k-fold cross-validation ($k=5$) on the valence prediction task ($N=60,000$).
>
> As shown in the table below, a linearly decodable valence representation emerges early (Layer 1) and remains stable, peaking in fidelity at Layer 3.
>
> **Table R1.1: Layer-wise Linear Probe Accuracy for Valence Prediction**
>
> | Layer | Accuracy (Mean %) | 95% CI |
> | :--- | :--- | :--- |
> | 0 | 81.36 | [80.78, 81.78] |
> | 1 | 97.17 | [97.08, 97.37] |
> | 2 | 98.96 | [98.78, 99.07] |
> | **3** | **99.08** | **[99.03, 99.14]** |
> | 4 | 98.85 | [98.74, 99.02] |
> | 5 | 97.38 | [97.28, 97.51] |
>
> The tight confidence intervals suggest valence representation is clearly decodable from the model's internal state at Layer 3, rather than a spurious correlation or an artifact of a specific data split.
>
> **Steering Confidence.** As before, we intervened on the residual stream at Layer 3 ($x \leftarrow x + \alpha \hat{w}$) and measured the mean logit shift for bond tokens. Confidence intervals were computed via bootstrapping ($N=1000$).
>
> **Table R1.2: Effect of Valence Steering ($\alpha$) on Bond Token Logits**
>
> | $\alpha$ | $\Delta$ Logit Single (`-`) | $\Delta$ Logit Double (`=`) | $\Delta$ Logit Triple (`#`) |
> | :--- | :--- | :--- | :--- |
> | **-2.0** | +0.033 [0.029, 0.036] | -0.019 [-0.024, -0.015] | -0.053 [-0.056, -0.049] |
> | **-1.0** | +0.014 [0.013, 0.016] | -0.008 [-0.011, -0.006] | -0.027 [-0.029, -0.026] |
> | **-0.5** | +0.008 [0.007, 0.009] | 0.000 [-0.001, 0.001] | -0.011 [-0.012, -0.010] |
> | **0.5** | -0.005 [-0.006, -0.004] | +0.002 [0.001, 0.004] | +0.008 [0.007, 0.009] |
> | **1.0** | -0.009 [-0.010, -0.007] | +0.009 [0.007, 0.011] | +0.022 [0.020, 0.023] |
> | **2.0** | -0.007 [-0.010, -0.004] | +0.017 [0.014, 0.021] | **+0.047 [0.044, 0.051]** |
>
>
> * **Positive Steering ($\alpha > 0$):** Simulating "high valence capacity" consistently suppresses single bonds (`-`) and promotes higher-order bonds (`=`, `#`). At $\alpha=2.0$, the probability mass shifts towards triple bonds ($\Delta \text{logit} \approx +0.047$), consistent with the chemical intuition that high valence budget permits higher bond orders.
> * **Negative Steering ($\alpha < 0$):** Simulating "low valence capacity" has the inverse effect, boosting single bonds while strongly suppressing triple bonds ($\Delta \text{logit} \approx -0.053$ at $\alpha=-2.0$), forcing the model towards lower-order connections.
>
> **Syntax Circuits.** We include a new experiment to see whether removing the heads that identify openers or significantly impact closer logit predictions would affect validity. We ablate these heads one-by-one and generate $10,000$ molecules with the edited model.
>
> **Table R1.3: Syntax Circuits**
> | Circuit | Head | Role | Pointer Mass | Causal ES | Ablation Validity |
> | :--- | :--- | :--- | :--- | :--- | :--- |
> | **Ring Closure** | L2H7 | Pointer | 0.307 | 0.51 | 79.6% |
> | | L1H2 | Writer | 0.033 | 4.98 | **25.4%** |
> | **Branching** | L2H3 | Hybrid | 0.490 | 0.58 | 63.7% |
> | **Control** | Avg. | Random | – | – | 90.3% |
>
> The results show that ablating the L1H2 ring closure head has a catastrophic impact on validity, while ring pointer ablation only causes a mild drop. The single branching head that we previously found important also impacts validity, though the effect is less dramatic than with L1H2. When compared with random attention heads, it's clear that L1H2 and L2H3 are critical to ensuring validity, while the model can survive ablating the average L2-L5 head.
>
> We have also added new experiments to assess SAE robustness, including an analysis of how SAEs behave when SMILES are randomized, and including an analysis of feature overlap between SAEs with different dictionary sizes.
>
> ### Conclusion
>
> We thank you again for your encouraging review and for highlighting the importance of quantifying uncertainty in mechanistic claims. We hope our response and the manuscript revision addressed your remaining concerns. We are happy to answer any remaining questions you might have.
>
>
> Best Regards, The Authors

---

> > ### Comment · Reviewer_SzMd · 2025-11-25
> >
> > I appreciate the efforts that the authors have made to improve the paper. I think is a solid contribution and therefore will continue recommending its acceptance.

---

### Author Response · Authors · 2025-11-29
**Revision Summary From Authors (1/2)**

We thank all reviewers for the detailed and constructive feedback. We have revised the manuscript, added new experiments, and clarified the positioning of our work to accommodate reviewer requests and solidify the contributions of this work. As a number of comments have overlapping themes, we deem it appropriate to address these concerns in one place and provide a brief summary for readers.

### TL;DR
Our work provides a mechanistic analysis of autoregressive transformers trained on drug-like small molecules. Reviewers have identified **(1)** statistical robustness, **(2)** anecdotal sparse autoencoder (SAE) analysis, **(3)** benchmark uncertainty, and **(4)** lack of clear practical utility as the main weaknesses of our work. We have addressed these concerns by **(1)** extending reporting on statistical significance and adding new ablations, **(2)** assessing SAE feature specificity, universality, and robustness, **(3)** refocusing evaluation around experimentally-backed benchmarks and including a number of deep learning baselines, **(4)** conducting steering experiments with sparse features and outlining directions for future work more relevant to practical applications.

These improvements, among others, are included in the revised manuscript, which now also contains extended reporting on experimental details and design choices, and provides additional data and figures for clarity. We refined the text to provide a more compelling argument for interpreting molecular transformers, as well as to clarify the value proposition of our work.

### 1. Circuit Robustness

Reviewers SzMd and UkWj have asked for stronger statistical treatment of mechanistic analyses for fundamental aspects of SMILES validity.

* **Syntax Circuits and Validity (Section 3.1)**
  We performed **head ablations** and measured the effect on SMILES validity to assess whether the identified circuits affect coherence during transformer inference. Results are summarized in the revised text and Table 1, making it clearer which heads are actually load-bearing.

  * We identify an attention head that consistently attends to ring openers but whose ablation only modestly reduces validity. A ring “writer” head (L1H2) is also identified whose ablation is catastrophic (validity drops to ~25% vs. ~90% for random heads).
  * A branch-balancing head (L2H3) is identified, whose ablation causes a substantial, but less catastrophic validity drop.

* **Valence Probes and Steering (Section 3.2)**
  We calculated 95% confidence intervals for layer-wise valence probe accuracy using stratified cross-validation, and for steering experiments using bootstrap resampling. The revised Figure 2 and associated text show that:

  * Valence capacity is linearly decodable (peak accuracy ≈ 99% at Layer 3).
  * Intervening along the valence direction produces a **monotonic, statistically significant** shift in bond-order logits.


### 2. SAE Robustness, Universality, Interpretability

Reviewers Qpvr and UkWj raised concerns about how stable and interpretable SAE features are, and whether they go beyond interesting anecdotes.

* **SMILES Augmentation Robustness (Section 4.1)**
  We added a new experiment comparing SAE activations on **randomized (non-canonical) SMILES** and canonical strings:

  * Early layers (L0–L1) show **high Jaccard and cosine similarity**, indicating that SAE features in these layers are robust to traversal order and align with local chemistry (atoms/bonds).
  * Deeper layers show reduced stability, consistent with inference-oriented behavior in autoregressive networks. A subset of late-layer features remain coherent.

* **Feature Universality (Appendix E.2)**
  To check whether features are artifacts of a particular run, we trained SAEs with 4x, 8x, and 16x expansion factors and compared decoder vectors:

  * Layer 1 features are highly reproducible, with mean max-cosine similarity ≳ 0.94 and >80% strict recovery of 8x features in the 16x dictionary.
  * Deeper layers show high subspace alignment but less 1:1 recovery, similar to how feature splitting is described in natural language settings.

* **Fragment specificity (Section 4.2)**
  We provide a more comprehensive assessment of fragment-based SAE feature screening to go beyond one-off feature analysis. SAE neurons are compared with dense MLP and PCA baselines on specificity for a set of SMARTS patterns representing common substructures.

  * SAE dictionaries consistently provide **sharper, highly specific detectors** for many functional groups (e.g., nitriles, sulfonamides) than dense residuals or PCA.
  * Dense/PCA representations capture some frequent motifs (e.g., simple carbonyls), but these are less selective with respect to intra-fragment activity.

These additions address robustness and interpretability questions and quantify how often SAEs provide chemically meaningful features.

---

> ### Author Response · Authors · 2025-11-29
> **Revision Summary From Authors (2/2)**
>
> ### 3. Downstream Benchmarks
>
> Reviewers UkWj, wRsj, and Qpvr asked for a clearer picture of real-world use-cases where our findings can prove useful. Reviewers also requested stronger baselines and more reliable benchmarks for validation on downstream tasks.
>
> * **New benchmarks (Section 5.1)**
>   Following Reviewer Qpvr’s suggestion, we added two more challenging benchmarks to the main text and moved Therapeutics Data Commons results to the appendix:
>
>   1. **cADME panel from Fang et al. (2023)**
>
>      * We evaluate HLM, MDR1, RLM, and Solubility.
>      * SAE features (with a simple XGB head) **consistently outperform** ECFP and the FCFP4+rdMD baseline from the original paper, and narrow the gap to supervised graph models.
>   2. **MoleculeACE activity-cliff benchmark (van Tilborg et al., 2022)**
>
>      * SAE features substantially outperform dense transformer embeddings and are competitive with ECFP, though expert-engineered features remain SOTA.
>      * When compared to deep learning baselines from the original study, unsupervised SAE features match or exceed the listed supervised sequence/graph models on overall RMSE and cliff RMSE.
>
> * **Repositioning of TDC**
>
>   * The main text focuses on MoleculeACE and the cADME panel.
>   * TDC results and leaderboard comparisons (including Chemprop and MapLight-GNN) are reported in the appendix as additional context.
>
> * **Complementarity with strong baselines**
>   Across tasks, we observe that:
>
>   * SAE features **bridge a substantial fraction of the gap** between simple fingerprints and state-of-the-art supervised models, despite being learned in a fully unsupervised way without explicit chemical priors.
>   * Combining SAE and ECFP features often yields further gains, indicating that sparse transformer features capture information beyond hand-crafted fingerprints.
>
> ---
>
> ### 4. SAE Steering
>
> Reviewer Qpvr asked whether mechanistic units can be used for controlling sample characteristics. We conducted a study using SAEs to condition molecular transformers:
>
> * **SAE-based feature steering (Section 5.2 & Appendix F.4)**
>   We perform **activation editing** using the top-k of max-pooled SAE features extracted from reference molecules, injecting them back into the residual stream during sampling. We observe:
>
>   * Consistent increase in pairwise Tanimoto similarity among steered samples, showing that SAE features concentrate generation around more restricted regions of SMILES space.
>   * A significant increase in similarity between generated samples and the target for a number of conditioning molecules.
>   * While SAEs should not be considered out-of-the-box conditioning tools, our results demonstrate that high-quality sparse features can behave as an extra control surface for shaping exploration during high-throughput sampling, even with relatively crude approaches to activation editing.
>
>
> ### Miscellaneous Improvements
>
> * **Terminology:** Reviewer wRsj pointed out the inaccurate use of *"chemical reasoning"* when referring to the generation process of a chemical language model. In response, we replaced all instances of the term with phrases that characterize such processes more accurately and do not imply cognition.
>
> * **Contributions:** Reviewer wRsj noted the claim of direct practical utility and real-world applicability in the Conclusion section. We clarify that while downstream task performance and the newly added steering experiments make a case for practicality, these capabilities are yet to be fully realized and their in-depth treatment is outside the scope of this paper. We now dedicate **Appendix A.2** to suggestions on future work for improving applicability in research and industrial contexts.
>
> - **Related Work:** Following Reviewer Qpvr’s suggestion, we now explicitly discuss previous work on molecular interpretability in the main paper and describe how our work relates to and extends these findings. MolGPT already observes syntactic/valence-related attention patterns via saliency. Our contribution is to move from one-off attention patterns to a more **mechanistic** analysis in search of discrete units of representation. We distinguish syntactic heads via targeted ablations, quantify causal impact on validity, and connect these circuits to linear valence features and sparse autoencoder (SAE) dictionaries. We also found it appropriate to extend the Related Work section in **Appendix A.1** to include relevant interpretability results from other drug modalities and molecules of life.
>
> ---
>
> ### Conclusion
>
> We are grateful for the reviewers’ detailed critiques, which substantially improved the clarity, scope, and empirical foundation of the paper. We hope this summary makes it easier to see how the revised manuscript addresses overlapping concerns about novelty, robustness, practical value, and positioning relative to prior work.
>
> Best regards, The Authors

---

### Meta-Review · Area_Chair_UZGM · 2026-01-17

**Summary:**

This submission presents a mechanistic interpretability analysis of an autoregressive transformer with a decoder-only architecture trained on canonical SMILES. It identifies attention head circuits associated with SMILES syntax, including ring closures and branch parentheses, a linearly decodable residual stream direction related to valence capacity that causally shifts bond token logits under intervention, and sparse autoencoder (SAE) features that align with chemically meaningful motifs. The revised version also evaluates SAE features on downstream property prediction benchmarks and reports preliminary inference time feature steering using SAE latents.

The initial review set included one strong accept from reviewer SzMd and several borderline below threshold reviews from reviewers UkWj, wRsj, and Qpvr, with recurring concerns about statistical rigor for causal claims, limited actionability and practical impact, positioning relative to prior work such as MolGPT, and overly strong terminology, especially the phrase chemical reasoning.

**Reviewer Concerns:**

Reviewer SzMd (R1): Addressed. Concerns regarding statistical robustness and uncertainty reporting were resolved through the addition of bootstrap confidence intervals and stratified cross-validation for probing and intervention results. R1 explicitly confirmed satisfaction with these improvements in the post-rebuttal discussion.

Reviewer UkWj (R2): Partially addressed. The authors added expressive GNN baselines and a steering demonstration to move beyond purely descriptive findings. However, the actionability of these mechanisms remains preliminary, and the absence of post-rebuttal feedback from UkWj leaves the resolution of these utility concerns unverified.

Reviewer wRsj (R3): Partially addressed. While "chemical reasoning" terminology was removed and causal evidence for syntax heads was strengthened via ablation, the core objection regarding the conceptual significance of SMILES syntax remains. A fundamental disagreement persists as to whether these findings provide the non-trivial chemical insights requested by the reviewer.

Reviewer Qpvr (R4): Largely addressed. The authors fulfilled specific conditional requirements by de-emphasizing TDC in favor of the MoleculeACE benchmark and clarifying the differentiation from MolGPT. These revisions align precisely with the reviewer’s stated roadmap for a positive recommendation.

**Reviewer Scores:**

Reviewer SzMd (R1): Unlikely to change score. The reviewer explicitly confirmed that the addition of statistical dispersion metrics and bootstrap confidence intervals addressed their primary concerns and maintained their strong support for acceptance.

Reviewer UkWj (R2): Unlikely to change score. While the authors added GNN baselines and steering experiments, the reviewer did not participate in the post-rebuttal discussion to acknowledge these changes, leaving the core concern regarding practical utility unverified.

Reviewer wRsj (R3): Unlikely to change score. Despite the removal of "chemical reasoning" terminology, the reviewer’s high-confidence conceptual objection regarding the significance of SMILES syntax analysis remains unaddressed by the added technical data.

Reviewer Qpvr (R4): Likely to increase score. The authors strictly followed the reviewer's conditional roadmap by de-emphasizing TDC in favor of more robust benchmarks and clarifying the connection to MolGPT, directly meeting the stated requirements for a positive score adjustment.

---

### Decision · Program_Chairs · 2026-01-26

Reject